

# Exploring future water resources and uses considering water demand scenarios and climate change for the French Sèvre Nantaise basin

Léonard Santos[1], Anthony Thomas[2], Gaëlle Tallec[1], Laurent Mounereau[2], Aaron Bluche[1], Bruno Lemaire[1,3], Rania Louafi[1], Guillaume Thirel[1,4]

[1]Université Paris-Saclay, INRAE, HYCAR Research Unit, Antony, France
[2]Etablissement Public Territorial du Bassin de la Sèvre Nantaise, Clisson, France
[3]Université Paris-Saclay, AgroParisTech, 91120, Palaiseau, France
[4]Univ Toulouse, CNES, CNRS, INRAE, IRD, CESBIO, Toulouse, France
*Correspondence to*: Guillaume Thirel (guillaume.thirel@inrae.fr)

**Abstract.**

Incorporating human influences within water resources modelling, in the context of global change, has proven to be a fruitful approach for improving the assessment of the impact of climate and water demand changes on hydrology and water demand satisfaction. In this study, we use an integrated water resources management model that details water withdrawal for irrigation,

drinking water supply, cattle watering and industry, as well as releases from wastewater treatment plants, drinking water network leakages, and industrial activities, within a catchment subject to human influence. Using this modelling approach and collaborating with relevant stakeholders through workshops, we developed a series of future water demand scenarios to examine the sustainability of water use in the future. Our findings indicate that climate change will be the primary driver of changes in water resources and water demand satisfaction. Moreover, we found that low flows and water demand satisfaction

will greatly decline in the future. A single climate projection indicates a less drastic deterioration of the system in certain areas of the catchment. We found that adapting water uses could help mitigate the negative impacts, though it is not fully satisfactory. The irrigation sector is set to be the most impacted in terms of water demand satisfaction. The study presents a methodological framework that helps to provide water sector managers with tailor-made results to support the design of effective adaptation measures.

## 1 Introduction

It is well established that climate change, characterized by rising air temperatures and altered precipitation patterns, significantly impacts water resources (Gleick, 1989; Arnell, 1999). Recent studies have enhanced our understanding of future streamflow evolution across various regions, including the Mediterranean basin (Noto et al., 2023), South America (Brêda et al., 2020), and Great Britain (Pastén-Zapata et al., 2020), among others.

While this phenomenon is already a significant environmental concern due to the minimal flows needed to sustain aquatic life, human water uses further exacerbate the issue. In many regions, the water cycle is heavily altered by water withdrawals and releases (Dong et al., 2022). Anthropogenic water withdrawals can be categorised into several sectors, including navigation (Salam et al., 2022), irrigation (Cheng et al., 2021), drinking water (Yalvaç et al., 2023), industry (Li et al., 2024), cattle



watering (Yang et al., 2024), among others. These diverse water uses naturally lead to a range of impacts on the water cycle.

These impacts can differ based on the timing of water withdrawals (i.e. rather summer for irrigation, and year-round for drinking water and industry) and the amount of water actually consumed. While the majority of water withdrawn for drinking and industrial uses is typically returned to the system, a significant proportion of water withdrawn for irrigation is consumed through processes such as evapotranspiration and plant growth (Starr and Levison, 2014). Additionally, water can be transferred within or outside of catchments or stored in large reservoirs to meet demand, resulting in significant alterations to

hydrological regimes (Peñas and Barquin, 2019).

It is evident that human interventions in water systems add complexity inherent to the natural dynamics of a catchment. As highlighted by the seminal work of Blöschl et al. (2019), a large proportion of the 23 Unsolved Problems in Hydrology relate to catchment-scale hydrological processes, reflecting the gaps in our understanding. The key challenge lies in the coexistence of numerous hydrological models, each based on different assumptions, without a clear frontrunner for accurately representing

hydrological processes (Hrachowitz and Clark, 2017). This highlights the need for continued research in hydrological modelling, an area that remains a focus of contemporary studies (Zhu et al., 2020; Tyralis et al., 2023). There is also growing research on hydrological processes understanding (Bracken et al., 2020; Dugdale et al., 2022).

However, when attempting to comprehend and depict anthropised catchments, it becomes clear that relying solely on hydrological models is insufficient. A range of approaches exists for representing both water resources and water uses (e.g.

Fard and Sarjoughian, 2021). The first approach involves separating water resources from water uses, followed by a posteriori comparison (Voisin et al., 2013). This approach aims to compare water availability with demand, identifying areas at risks of water shortage or scarcity. This approach is the simplest, given the availability of many hydrological models and direct mobilization of observation data. However, obtaining accurate water use data can be challenging, though several databases are available for investigating annual water uses (e.g., Lopez et al., 2024). These databases indeed cover limited areas, periods, or

specific water uses. Additionally, water use can be assessed using coarse models (Nazemi and Wheater, 2015).

A second approach involves integrated water resources modelling, which combines hydrological models with water demand models. In this context, water demand satisfaction is constrained by fine scale temporal and spatial water availability, and streamflow is directly influenced by withdrawals or releases. Such models typically include a component that simulates water management practices, such as imposing water withdrawal restrictions during low flows or outlining the operational principles

of dams and reservoirs (Lemaitre-Basset et al., 2024).

In the era of global change, the ability to represent the combined evolution of hydrology and water uses is of paramount importance for water managers (Carmona et al., 2013; Voisin et al., 2017). To anticipate how both resources and uses will evolve, and how conflicts may arise due to water shortages, these models need to be applied in conjunction with climate projections (Hadri et al., 2022) and water demand scenarios. Evaluating adaptation strategies is key to assessing the

sustainability of water use, i.e. examining how future modifications in water use could improve or worsen the situation for both natural resources and water demand satisfaction (Lemaitre-Basset et al., 2024).





In France, the Loire-Bretagne Water Agency has identified integrated and sustainable water resource management as a key concern. In order to address this concern, the Agency provides funding for studies focusing on hydrology, environment, uses and climate (HMUC, for Hydrologie, Milieux, Usages et Climat). The objective of these studies is to formulate a local water management strategy with the aim of restoring the quantitative balance of water resources and enhancing local knowledge (hydrogeological functioning of the basin, inventory of withdrawals and discharges) in order to identify the causes of malfunctioning and assess the availability of water resources. In addition to extensive work on the collection of hydroclimatic and anthropogenic use data, as well as consultation with local stakeholders to determine future trends in water usage, these studies aim to combine the four components of hydrology, environment, uses and climate. In view of the restricted availability of human, financial and technical resources, a large proportion of these studies has been confined to cross-referencing these aspects through a posteriori comparison (EPTB Vienne, 2024; Syndicat Grand Lieu Estuaire, 2024). This corresponds to the first and simplest of the modelling options described above, and therefore involves numerous simplifications. Conversely, certain HMUC studies, albeit less prevalent, advocate for the direct incorporation of water uses in hydrological modelling (Etablissement Public Loire, 2024). However, this approach remains unelucidated within the reports and is not substantiated by scientific publications. The incorporation of water uses in hydrological modelling has been extensively implemented in the Sèvre Nantaise catchment area in western France (Santos et al., 2023b).The central objective of this study is twofold: i) to describe an integrated water resources modelling chain comprising a daily hydrological model, the use of climate input data, the use of water withdrawal and release models and the development of future water demand scenarios, and ii) to question the sustainability of water uses in a heavily modified catchment, in order to assess the potential added value of adaptation strategies using water use scenarios, for both hydrology and water use satisfaction. The sub-objectives of this study are: first, to assess the impact of climate change on the natural (uninfluenced) hydrology of the Sèvre Nantaise; second, to evaluate the impact of climate change on the influenced hydrology and on the water demand satisfaction; and third, to explore the impact of water demand scenarios on the hydrology and on the water demand satisfaction.

## 2 Materials and methods

### 2.1 Study area: the Sèvre Nantaise catchment

#### 2.1.1 Geophysical description of the Sèvre Nantaise catchment

The Sèvre Nantaise is the most downstream tributary of the France's longest river, the Loire. It flows into the Loire River at the city of Nantes. The catchment has a gentle relief, with the highest point reaching an altitude of 215 m. The river spans a total length of 136 km and the basin area is 2,350 km². Its main tributaries include the Maine on the left bank, who take streamflow from the Petite Maine and the Grande Maine, as well as the Ouine, the Ouin, the Moine and the Sanguèze on the right bank (Figure 1).





The predominant land uses are agricultural (88.2 %), followed by urban (8 %) and natural (3.5 %) areas. Urban areas are concentrated around one main city, Nantes, and several medium-sized cities (between 5,000 and 55,000 inhabitants), including Cholet and Clisson, amongst others. Open water constitutes the remaining surface area. The Sèvre Nantaise catchment is located within an area of granite bedrock that is composed of metamorphic, crystalline, and volcanic formations.

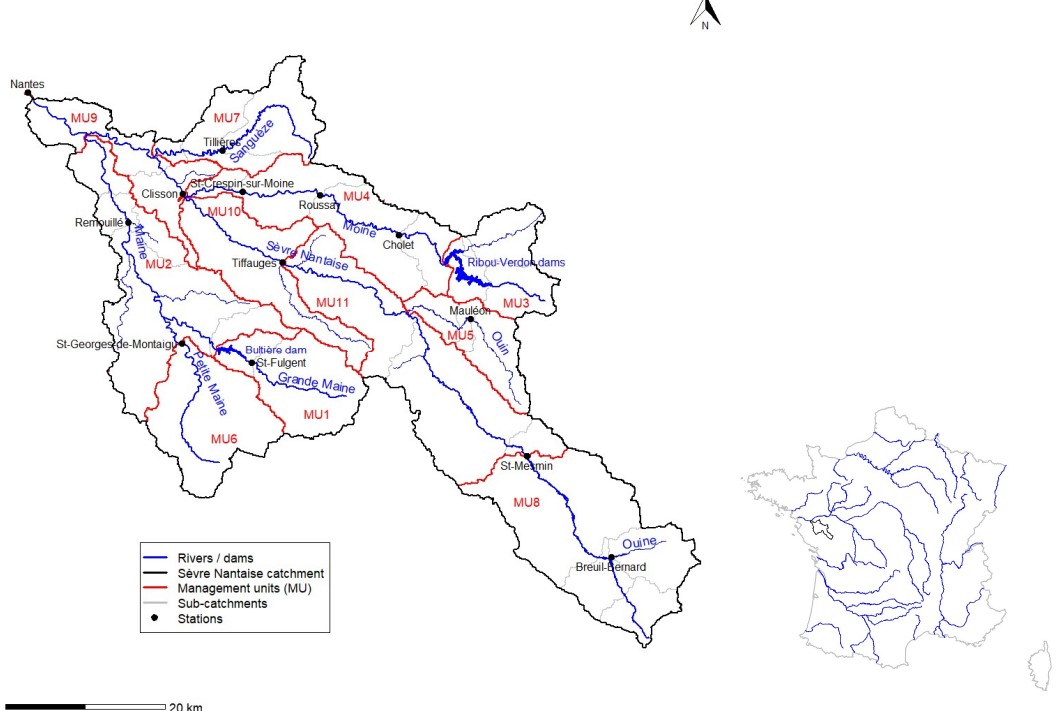

**Figure 1: Map of the Sèvre Nantaise catchment (left) and location in France (right). Management units (MUs) represent spatial entities used for water resource planning; sub-catchments are sub-divisions needed for the hydrological model (see details in section 2.3.1); hydrometric stations, detailed in Table 2, are used for model calibration and evaluation**

### 2.1.2    Current hydrometeorological conditions

Due to its proximity to the Atlantic Ocean, the study area has a temperate climate with oceanic influences, identified as Cfb in the Köppen-Geiger classification by Strohmenger et al. (2024). Based on climate data from the SAFRAN atmospheric reanalysis (Vidal et al., 2010), mean monthly air temperature ranges from 4 to 19 °C. The average annual temperature over the period 1958-2020 is about 10.5 °C upstream, and 11.5 °C downstream. Based on the observed data from the Nantes meteorological station, located close to the catchment outlet, the temperature increase is about 0.2 to 0.3 °C per decade. The annual potential evapotranspiration, calculated with the Penman-Monteith formulation (Allen et al., 1998), is slightly above





700 mm y$^{-1}$ over the period 1958-2020, and follows an increasing trend. Annual precipitation varies between 700 and 1000 mm y$^{-1}$ when averaged over the catchment. It is higher upstream (about 1000 mm y$^{-1}$) and lower downstream (about 700 mm y$^{-1}$) over 1958-2020. Monthly precipitation is relatively uneven, with winters (around 100 mm month$^{-1}$) wetter than summers (less than 50 mm month$^{-1}$). Based on the national HydroPortail archive (Dufeu et al., 2022), we can qualify the hydrological regime of the Sèvre Nantaise as pluvial (Sauquet et al., 2024). Due to the bedrock underground, groundwater storage is very low in the basin which, in addition to the pluvial regime, leads to a strong seasonality in streamflow. Streamflow is therefore higher in winter, and very low in summer. Some parts of the smaller tributaries can even be impacted by periods of no flow.

### 2.1.3 An anthropised catchment

The hydrology of the Sèvre Nantaise is influenced by several hydraulic structures designed to meet the water demands of various human activities, including irrigation, drinking water supply, cattle watering and industry. The Bultière reservoir, located on the main course of the Grande Maine River (Figure 1), has a maximum capacity of 5 10$^6$ m³ and supplies water to a wastewater treatment plant with a maximum daily production capacity of 22,000 m³. The Ribou-Verdon reservoirs, located on the Moine River, have a combined maximum capacity of 17.6 10$^6$ m³ and provide drinking water to a treatment plant with a maximum daily production capacity of 24,000 m³. In addition to these large reservoirs, there are approximatively 11,500 smaller reservoirs, with a total volume estimated at 54.9 Mm³ (Table 1). These smaller reservoirs (when used) serve various purposes, including agricultural and industrial water supply, as well as recreational activities and their influence on hydrology remains partially unknown. Furthermore, the operation of wastewater treatment plants within the catchment results in the release of water at multiple locations. Due to the area's geology, characterized by the absence of deep aquifers, most water abstraction occurs from surface water or shallow aquifers. In addition, water transfers between neighbouring catchments are possible. For example, about half of the drinking water is imported from outside of the Sèvre Nantaise catchment; conversely, a part of the Bultière reservoir water is used to provide drinking water to population living out from the Sèvre Nantaise basin. For a more comprehensive overview of the anthropogenic influences on the catchment, refer to Santos et al. (2022, 2023a). The following sections provide detailed data on water withdrawn and released volumes.

**Table 1: Number of small reservoirs on the Sèvre Nantaise catchment for three surface categories.**

| Surface category | Number of small reservoirs | Part of the total [%] | Total volume [Mm³] |
|---|---|---|---|
| S < 1000 m² | 6917 | 60.25 | 3.2 |
| 1000 m² < S < 10 000 m² | 4104 | 35.75 | 21.1 |
| S > 10 000 m² | 460 | 4.00 | 30.6 |



The regulatory framework for water management in the Sèvre Nantaise catchment defines the principle of balanced and
sustainable water resources. It establishes priorities for water allocation, with health, safety, and drinking water supply taking
precedence, followed by environmental needs and flood protection. In the event of a water shortage, local authorities are
empowered to impose a series of restrictions across sub-catchments. These measures are primarily based on streamflow
observations from predefined gauging stations, while also considering local conditions, which may vary over time and space.

**2.2  Data**

**2.2.1    Observed hydrometeorological data**

Climate data from the SAFRAN atmospheric reanalysis were used (Vidal et al., 2010). This reanalysis provides daily
precipitation, air temperature and potential evapotranspiration (PE) on an 8 x 8-km² regular grid. PE was estimated using a
modified version of the Penman-Monteith formulation (Allen et al., 1998), in which radiation data were replaced with air
temperature estimates following the Hargreaves formulation (Hargreaves and Samani, 1985). This modification ensures a
homogeneous estimation of PE, as some regional climate models do not account for evolutive aerosols, which can impact their
radiation estimation (Boé et al., 2020). Daily streamflow data were retrieved from the national HydroPortail database (Dufeu
et al., 2022). These datasets, which had previously been quality-checked by data providers, were subjected to further visual
inspection in order to detect erroneous data. A total of 13 hydrological stations were selected for the study (Table 2). The data
covered the 2008-2020 period, without any gap. It can be seen in Table 2 that the ratio between Q25 and Q75 can differ a lot
between very contrasted stations (e.g. on the Sanguèze) or less contrasted stations (e.g. on the downstream Sèvre Nantaise).

**Table 2: Hydrometric stations (see Figure 1 for location), on which the model is calibrated and evaluated, and their observed
characteristics. Q25 and Q75 are the 25th (i.e. exceeded 75 % of the time) and 75th (i.e. exceeded 25 % of the time) quantiles of
daily streamflow, respectively. The 2008-2020 period is used.**

| Station code | Name | River | Catchment area (km²) | Q25 streamflow (m³ s⁻¹) | Mean streamflow (m³ s⁻¹) | Q75 streamflow (m³ s⁻¹) |
|---|---|---|---|---|---|---|
| M700561010 | Breuil-Bernard | Ouine | 63 | 0.04 | 0.71 | 0.81 |
| M702241010 | St-Mesmin | Sèvre Nantaise | 364 | 0.48 | 4.24 | 4.89 |
| M704401010 | Mauléon | Ouin | 60 | 0.06 | 0.60 | 0.68 |
| M711241010 | Tiffauges | Sèvre Nantaise | 817 | 1.00 | 9.33 | 10.80 |
| M720302010 | Cholet | Moine | 176 | 0.45 | 1.31 | 1.27 |
| M721301010 | Roussay | Moine | 287 | 0.61 | 2.14 | 2.20 |



| M721302010 | St-Crespin-sur-Moine | Moine | 366 | 0.75 | 3.00 | 2.94 |
|---|---|---|---|---|---|---|
| M730242011 | Clisson | Sèvre Nantaise | 1381 | 1.85 | 13.90 | 15.80 |
| M731401010 | Tillières | Sanguèze | 93 | 0.01 | 0.76 | 0.62 |
| M741301010 | St-Fulgent | Grande Maine | 132 | 0.08 | 1.26 | 1.27 |
| M743311010 | St-Georges-de-Montaigu | Petite Maine | 192 | 0.06 | 1.66 | 1.41 |
| M745301010 | Remouillé | Maine | 595 | 0.31 | 5.24 | 4.58 |
| M7502410 | Nantes | Sèvre Nantaise | 2354 | 2.47 | 22.76 | 24.20 |

### 2.2.2 Climate projections

To assess the impact of climate change on hydrology and water use satisfaction, a classical modelling chain was employed. To this end, we used climate projections produced by the Explore2 project, funded by the Ministry of the Environment (Sauquet et al., 2025). The Explore2 dataset includes a selection of climate projections from 17 General Circulation Models (GCMs) /

Regional Climate Models (RCMs) couples extracted from the Euro-Cordex initiative (Jacob et al., 2014, 2020), corresponding to the CMIP5 experiment (Taylor et al., 2012). Given the large volume of available data (Marson et al., 2024), a subset of five GCM/RCM projections was selected, using the greenhouse gases emission scenario corresponding to the Representative Concentration Pathway 8.5 (RCP 8.5) and using the ADAMONT bias-correction method (Verfaillie et al., 2017). The selection of the five GCM/RCM projections followed the recommendations of Marson et al. (2024), based on the following criteria: i)

consistency with climate evolution as predicted by the newly released CMIP6 projections, which had not yet been regionalised over France for a sufficient number of models, ii) representativeness of contrasted precipitation and temperature evolutions for RCP 8.5 and the end of the XXI[st] century. The selection of a limited set of projections was made to avoid interpreting climate change impacts in a probabilistic manner. Instead, the objective was to propose example scenarios for catchment stakeholders to use in future planning. Therefore, results from the different projections will not be aggregated. A

comprehensive analysis of precipitation and temperature evolutions was conducted for RCP 8.5, focusing on the 2056–2085 period relative to 1976–2005. This analysis encompassed annual and seasonal variations across the catchment. The five selected projections are presented in Table 3, while Appendix A compares the actual evolution of precipitation and air temperature for the five projections across all available projections. As shown in Table 3 and Appendix A, all projections result to an increase in temperature year-round, particularly during summer and autumn. Furthermore, all projections indicate either

an increase or no change in winter precipitation and a decrease or no change in summer and autumn precipitation, with contrasted patterns for spring and the entire year.



**Table 3: List of selected climate projections and their characteristics in terms of precipitation and air temperature evolution for**
**2056-2085 relatively to the 1976-2005 period assessed in a qualitative way based on results from Appendix A.**

| Name | GCM | RCM | Precipitation evolution | | | | | Temperature evolution | | | | |
|---|---|---|---|---|---|---|---|---|---|---|---|---|
| | | | Year | DJF | MAM | JJA | SON | Year | DJF | MAM | JJA | SON |
| A1 | CNRM-CERFACS-CNRM-CM5 | CNRM-ALADIN63 | = | + | = | -- | = | + | + | ++ | + | ++ |
| B3 | ICHEC-EC-EARTH | MOHC-HadREM3-GA7-05 | - | = | = | -- | - | ++ | + | ++ | +++ | ++ |
| C1 | MOHC-HadGEM2-ES | CNRM-ALADIN63 | ++ | ++ | ++ | = | - | ++ | ++ | ++ | ++ | +++ |
| C2 | MOHC-HadGEM2-ES | CLMcom-CCLM4-8-17 | - | ++ | = | --- | -- | +++ | +++ | ++ | +++ | +++ |
| F4 | NCC-NorESM1-M | DMI-HIRHAM5 | - | + | - | -- | - | ++ | ++ | ++ | ++ | ++ |

### 2.2.3    Water withdrawal and release data

A close collaboration with stakeholders ensured the successful retrieval of water withdrawal and release data from national
and local water agency databases, as well as directly from water users. Most data series were available from 2008 to 2020.
While some data were available on a daily scale (particularly for large reservoirs), most were provided at coarser time intervals:
monthly for wastewater treatment plant releases, or annually for industrial withdrawals and releases, as well as irrigation and
cattle watering (although the latter was estimated knowing the number of heads). Regarding irrigation and drinking water
withdrawals, data at finer scales were accessible, but only for a limited number of years. Finally, some areas had missing data
because certain users did not provide their data.

Spatial and temporal extrapolation, as well as temporal disaggregation, were carried out and validated by local water
stakeholders from the Sèvre Nantaise catchment. After extensive analysis, a daily time step database was compiled for all the
water uses within the catchment area. However, this article does not focus on that aspect of the research. Readers are therefore
encouraged to refer to Santos et al. (2023a) for a more comprehensive description of the database extension. In the present
work, the water withdrawal and release data are considered as granted, though it is recognised that uncertainties exist, as is the
case with other observed data, such as climatic or hydrological data.



Figure 2 illustrates the evolution of annual water withdrawals across four sectors (drinking water, cattle watering, irrigation and industry) and annual water releases for three sectors (wastewater treatment plants, drinking water leakage and industry). It clearly shows that the volume of water withdrawn for drinking water supply, cattle watering and industry, remains relatively stable in time, whereas withdrawals for irrigation are more variable, likely due to their sensitivity to climatic variations. Regarding water releases, the volumes from industry and drinking water leakage remain stable over the period, while the

amount released by wastewater treatment plants exhibits greater variability. This variability is likely attributable to fluctuating precipitation levels that locally contribute to the release. Additionally, it is notable that total water withdrawals exceed water releases within the catchment by approximatively 35 % (Figure 2). This water withdrawal exceedance is mainly explained by irrigation and cattle watering even if it is partially compensated by drinking water importation from outside of the catchment. Substantial spatial heterogeneities in water withdrawn also exist within the catchment (Table 4). Specifically, drinking water

supply are concentrated in a few management units, including the two where the major dams are located, whereas releases from wastewater treatment plants and network leakage are scattered. Other water uses are also unevenly distributed over the Sèvre Nantaise catchment. In addition, the Ribou-Verdon dam aims at sustaining low flows, but the Bultière dam does not, and small reservoirs tend to delay the streamflow increase after the low-flow period.

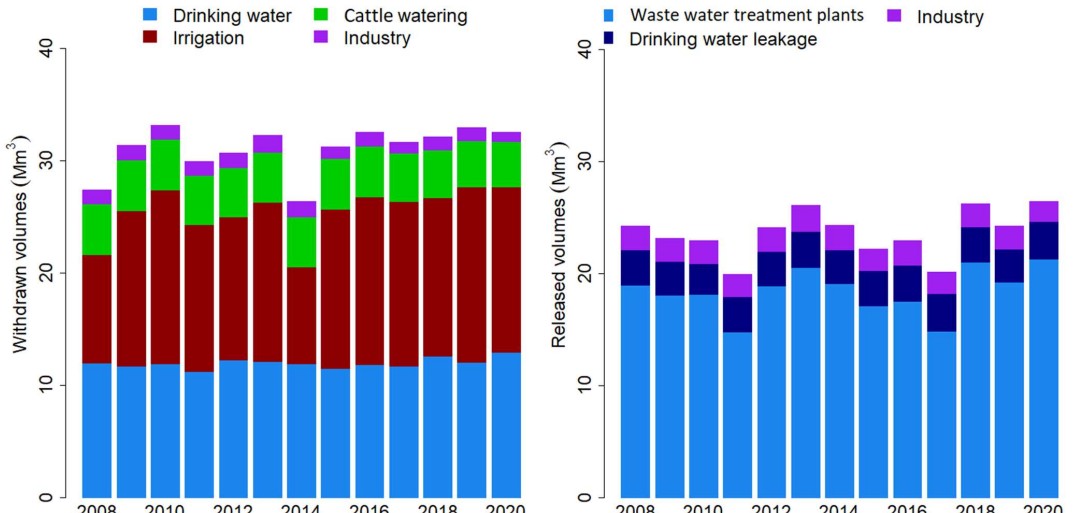

**Figure 2: Left: Evolution of annual water volumes withdrawn for drinking water, cattle watering (excluding water withdrawn from the drinking water supply), irrigation and industry. Right: Evolution of annual water volumes released by wastewater treatment plants, drinking water leakage and industry. All data are measured or interpolated from measurements (see Santos et al., 2023a).**





**Table 4: Management Units (MUs), on which water uses are discussed, their upstream areas, and their withdrawn and released**
**mean annual volumes over 2008-2020 for each water use. SN stands for Sèvre Nantaise and Ind. stands for Industries.**

| ID | MU Name | Catchment area (km²) | Mean annual withdrawals (Mm³) | | | | | Mean annual releases (Mm³) | | | |
|---|---|---|---|---|---|---|---|---|---|---|---|
| | | | Drinking water supply | Irrigation | Cattle watering | Ind. | Total | Wastewater treatment plants | Drinking water leakage | Ind. | Total |
| MU1 | Grande Maine | 159 | 5.34 | 0.93 | 0.36 | 0.01 | 6.64 | 1.39 | 0.22 | 0.01 | 1.62 |
| MU2 | Maine aval | 676 | - | 3.16 | 0.54 | 0.20 | 3.90 | 2.37 | 0.50 | 0.76 | 3.63 |
| MU3 | Moine amont | 133 | 4.69 | 0.79 | 0.25 | - | 5.73 | 0.50 | 0.07 | - | 0.57 |
| MU4 | Moine aval | 384 | - | 1.44 | 0.43 | 0.07 | 1.94 | 6.31 | 0.51 | 0.03 | 6.85 |
| MU5 | Ouin | 100 | 0.21 | 0.17 | 0.21 | 0.23 | 0.82 | 0.51 | 0.06 | 0.23 | 0.80 |
| MU6 | Petite Maine | 192 | - | 0.99 | 0.44 | 0.10 | 1.53 | 0.89 | 0.26 | 0.15 | 1.30 |
| MU7 | Sanguèze | 161 | - | 0.35 | 0.20 | - | 0.55 | 1.02 | 0.14 | 0.27 | 1.43 |
| MU8 | SN amont | 364 | - | 0.94 | 0.71 | - | 1.65 | 0.50 | 0.14 | - | 0.64 |
| MU9 | SN aval | 2354 | - | 0.33 | 0.04 | 0.58 | 0.95 | 1.84 | 0.76 | 0.58 | 3.18 |
| MU10 | SN Clisson | 1381 | - | 2.13 | 0.37 | - | 2.50 | 1.05 | 0.14 | 0.01 | 1.20 |
| MU11 | SN moyenne | 817 | 1.73 | 2.30 | 0.85 | - | 4.88 | 2.05 | 0.33 | - | 2.38 |
| Whole catchment | | | 11.97 | 13.53 | 4.40 | 1.19 | 31.09 | 18.43 | 3.13 | 2.04 | 23.60 |

## 2.3 Hydrological modelling including integrated water resources management

### 2.3.1 The rainfall-runoff model used for representing the natural hydrological cycle

River streamflow is simulated with the GR6J hydrological model, using a semi-distributed approach. The GR6J model is a
daily lumped process-based hydrological model that was developed for low-flow simulation (Pushpalatha et al., 2011; Tilmant
et al., 2020). It is a modified version of the GR4J model (Perrin et al., 2003). The GR6J model is a bucket-type model with six
parameters requiring calibration, X1 to X6. X1 and X3 are the production and the routing store capacity parameters (mm),
respectively. X4 is the unit hydrograph time base (in days). X2 and X5 are the inter-catchment exchange coefficient (mm d$^{-1}$)
and the exchange threshold (unitless), respectively. X6 is the exponential store capacity parameter (mm).
The study catchment is subdivided into 32 sub-catchments on which the GR6J model is applied in a semi-distributed manner.
This approach utilises specific climatic inputs (precipitation and potential evapotranspiration) along with parameter values for
each sub-catchment to simulate the corresponding streamflow. The upstream streamflow is then routed to the downstream sub-
catchment using a lag function and an additional parameter L (m s$^{-1}$), which represents the in-stream celerity (Lobligeois et





al., 2014). Semi-distributed versions of GR hydrological models have been successfully applied in the context of climate
change (Sauquet et al., 2025; Thirel et al., 2025) and influenced hydrology (Lemaitre-Basset et al., 2024). The subdivision of
the catchment into 32 sub-catchments was designed to align with locations of the 13 gauge stations (listed in Table 2) and the
11 management units (MUs) used for water management in the Sèvre Nantaise (Figure 1). For sub-catchments without a
gauging station at their outlet, GR6J parameters are transferred from upstream catchments, as proposed by de Lavenne et al.
(2019).

**2.3.2   Integrated water resources management modelling**

To accurately represent the hydrological processes of the Sèvre Nantaise catchment, which are influenced by water
withdrawals and releases, an integrated water resources management modelling approach was employed. This entails
modelling natural water resources (i.e. streamflow) while incorporating withdrawals, releases and management practices of
water (e.g. restrictions). The open source "airGRiwrm" R package was used for this purpose (Dorchies et al., 2023). This
package facilitates the establishment of hydrological modelling with semi-distributed GR models, leveraging the capabilities
of the airGR R package (Coron et al., 2017, 2020), which provides the GR hydrological models, including GR6J. Beyond the
functionalities of airGR, the airGRiwrm package automates the semi-distribution of the GR models and integrates human water
uses (e.g. withdrawals, releases, and dams), as well as water management practices. In this case, human water uses can be
derived either from observed time series or from water demand models. This tool ensures that the human-influenced part of
the water cycle is taken into account by water demand models (or their observed time series) and water restriction rules, while
the natural water cycle is represented through the rainfall-runoff model. The model parameters thus describe a more natural
rainfall-runoff relationship.

Several water demand models were implemented, namely for drinking water, irrigation, cattle watering and industry. In
addition, water release models were implemented for industry, drinking water and wastewater treatment. All these models are
necessary to estimate future water use fluxes. Their equations are detailed, in addition to the Ribou-Verdon and Bultière dams,
in Appendix B. In essence, the cattle water demand is proportional to the number of heads, the respective consumption per
head and is partly withdrawn from the drinking water network or from the natural environment and distributed over the year.
The industrial water demand is estimated based on the consumption from each industry, and is partly withdrawn from the
drinking water network or from the natural environment and distributed over the year. The returning water for industry is partly
released to the wastewater treatment network and to the natural environment. The drinking water withdrawal is proportional
to the number of inhabitants, to the unit consumption, and to the losses of networks, whereas the release is proportional to the
percentage of habitations linked to the wastewater treatment plants, and to the withdrawals. Regarding irrigation, the CropWat
(Allen et al., 1998) irrigation demand described in Soutif-Bellenger et al. (2023) is used. The demand for irrigation water is
determined by various factors, including the type of crop, the availability of a small reservoir, the percentage of irrigated area,
and the prevailing climate conditions. This demand is then met through the withdrawal of water from either the natural
environment or small reservoirs.



In addition to the model's fluxes and stocks, specific resource management rules (i.e. restrictions) must be implemented during periods of drought. Indeed, when the water resources are insufficient to satisfy all the uses simultaneously, which is estimated through the use of streamflow thresholds, the model prescribes restrictions with different levels. The first level of restriction

leads to an interdiction of irrigation as well as a 25 % reduction of industrial withdrawals outside the drinking water network. At the second level of restriction, irrigation and industry withdrawals are stopped, and the drinking water withdrawals are reduced by 5 %. In circumstances where streamflow is at its lowest, the model implements an order of priority for the uses, emulating the real-world practices, as follows: drinking water production, cattle watering, industrial production, irrigation, and finally, small reservoirs filling up. For each sub-catchment, the aforementioned uses are satisfied in this order.

**2.3.3    Hydrological modelling set-up**

In the following, we use the model under three configurations (Figure 3): i) the model that incorporates observed withdrawals and releases (i.e. the calibrated model, hereafter referred to as "Calib"), ii) the model that excludes water uses to assess their impact on the Sèvre Nantaise hydrology and the projected evolution of hydrology without taking water uses into account (the uninfluenced model, hereafter referred to as "Uninf"), and iii) the model where water uses are determined from water demand

and release models, which will be used alongside climate projections and water use scenarios (the integrated water resources management model, hereafter referred to as "iwrm"). Note that the GR6J model in the "Uninf" and "iwrm" models was not recalibrated and use the "Calib" parameter values.

The "Calib" model will mainly be used for optimising the GR6J model parameters. The three models, forced by SAFRAN over 2008-2020, will be used to compare the simulated streamflow to observed streamflow, highlighting the impact of water

uses on streamflow, and helping to validate the water demand models. The "Uninf" model will also be used to provide an assessment of the evolution of natural streamflow under climate change. Finally, the "iwrm" model will also be used to assess the impact of climate change on influenced streamflow, water demand, and water demand satisfaction. These two last applications will necessitate the use of climate projections, and, for "iwrm", of water use scenarios.



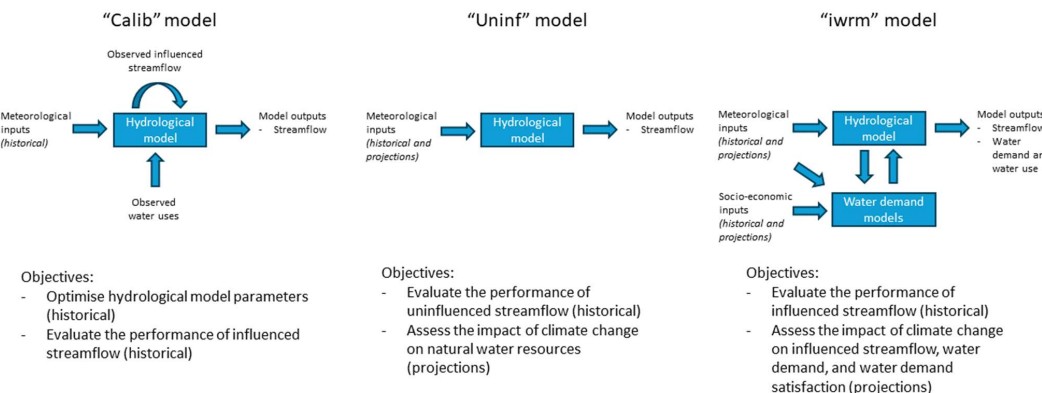

**Figure 3: The three conceptual hydrological modelling frameworks integrating, or not, water uses, and their objectives.**

Before using the model for prospective objectives, we first calibrated it ("Calib" model) against measured, therefore influenced, streamflow values. Based on the work of Santos et al. (2018) and Thirel et al. (2024), we applied the KGE criterion (Gupta et al., 2009) with a Box–Cox transformation of streamflow, which normalizes the streamflow distribution, giving a similar weight to high and low flows. The model parameters were optimised sequentially, i.e. from upstream catchments to downstream catchments, using a version of the model in which the observed water releases and withdrawals described in section 2.2.3 were incorporated. Parameter optimisation was based on streamflow data, observed between 2008 and 2020, the period for which water use data were available, with an initialization of the model over the 2006 and 2007 years. To ensure that the parameters are not overly specific to this time period, we conducted a calibration-control test (i.e. a verification of parameter performance and values over an independent period), as advised by Klemeš (1986). The data period was divided into two equal 6-year sub-periods (Oct. 2008-Sep. 2014 and Oct. 2014-Sep. 2020), which were used for two separate optimisations. Performance was then evaluated over these sub-periods to check that performance degradation, outside the optimisation period, was low. The results showed that performance losses remain limited, suggesting that the selected parameters are applicable for future periods (not shown here).

### 2.3.4 Hydrological model evaluation

In this article, we present the model performance for the period from 2008 to 2020. The performance is assessed using the KGE criterion (Gupta et al., 2009) along with its components for bias and correlation. To place more emphasis on low flows, we also provide the KGE criterion value after applying a Box–Cox transformation. To evaluate the impact of climate change and water uses on streamflow, we use a set of hydrological indicators that scan different segments of the streamflow range. Specially, we focus on the 5th (low flow), 50th (mid flow) an 90th (high flow) quantiles, as well as the average streamflow (QA)





and the mean annual minimum monthly streamflow with a 5-year return period (noted QMNA5 and widely used in French water management regulation).

## 2.4 Exploratory scenarios for projections of water demands

Since the modelling framework explicitly includes water demand models and the mechanisms for meeting this demand through water management rules, it allows to consider the water demand evolution at the catchment scale, in conjunction with the impact of climate change on water resources availability. To cover a wide range of potential water demand changes, three distinct scenarios were proposed, each adapted to the different water use sectors, through a three-step process:

- First, national to regional reports describing adaptation plans were reviewed to create an initial set of three scenarios,
each representing a possible evolution of water demand for every water use sector.
- Second, these three scenarios were presented to the local stakeholders (agricultural advisors, drinking water and wastewater treatment companies, environmental representatives, and local and regional officials) during a workshop in which we reviewed and discussed each hypothesis and the corresponding water demand evolution.
- Third, modifications of the water demand projections were made based on feedback from the stakeholders, who then
validated the revised scenarios.

The first scenario assumes a constant water demand. The second scenario reflects future water demand following recent trends, while the third scenario presents an alternative future, with large changes in demand. Namely, for the alternative scenario, higher efforts are being made to improve the efficiency of the drinking water network and industrial consumption per unit, there is a decrease in the number of cattle heads, and a greater proportion of water withdrawal for cattle coming from the

drinking water network, and there is an halt of the decrease in the total cultivated area as well as a modification of crop rotation. The details of the three scenarios are presented in Appendix C and in Santos et al. (2023b).

## 3 Results

### 3.1 Performance of the hydrological models

Table 5 shows that the models perform reasonably well when forced by SAFRAN over the 2008-2020 period, with most KGE

values above 0.8, with only few exceptions. The Tillières station shows the lowest KGE values across all three models. For most stations, the KGE values are rather similar across the three models, with differences of less than 0.02. This means that the water releases and withdrawals have a limited impact on the overall streamflow regime at most stations. Exceptions are observed on the Moine River, where the Ribou-Verdon dams are located. Specially, the upstream Cholet station (where the calibrated model performs best) and the more downstream Roussay and St-Crespin-sur-Moine stations (where the integrated

model performs the worst) show differing results. These findings indicate that the Ribou-Verdon dams impact a lot the streamflow regime, and their representation in the iwrm model may be imperfect. Looking at the bias score, we observe that streamflow is underestimated when the dam is not neglected or represented by a model. However, this issue is somewhat



mitigated when examining the KGE Box–Cox values. Indeed, this criterion shows that the iwrm model outperforms the uninfluenced model for almost all stations, including those on the Moine River. This means that the iwrm model performs well

for low flows, which is important for the representation of water resources and the evaluation of water use satisfaction under climate change. Finally, Table 5 reveals that the correlation is relatively similar between the three models for all stations.

These observations are further supported by Figure 4, where the observed interannual streamflow regimes are very well reproduced by all three model versions at most gauge stations. Discrepancies between the models are most notable on the Moine River (Cholet, Roussay, St-Crespin-sur-Moine), where low flows are underestimated by the uninfluenced model, and

high flows are underestimated by all the models even the one incorporating observed influences. All models also show slightly weaker performance at the Sanguèze (Tillières) station.

**Table 5: Performance criteria for the three model versions forced by SAFRAN for 2008-2020 for the 13 gauge stations. For each criterion, the score for the best model is shown in bold, and the score for the worst model is shown in italics. The KGE, correlation**
**and KGE box cox criteria are negatively oriented and the best value is 1. The Bias best value is 1, and a value above 1 depicts an overestimation, while a value below 1 depicts an underestimation.**

| Criterion | KGE | | | Bias | | | Correlation | | | KGE box cox | | |
|---|---|---|---|---|---|---|---|---|---|---|---|---|
| Hydrological station \ model | Calib | Uninf | iwrm | Calib | Uninf | iwrm | Calib | Uninf | iwrm | Calib | Uninf | iwrm |
| Breuil-Bernard | *0.86* | **0.87** | **0.87** | *0.96* | 1.01 | **1.00** | **0.91** | **0.91** | **0.91** | 0.91 | *0.90* | **0.93** |
| St-Mesmin | *0.85* | *0.85* | **0.86** | *0.97* | **0.99** | **0.99** | **0.93** | **0.93** | **0.93** | **0.97** | *0.93* | 0.96 |
| Mauléon | *0.81* | *0.81* | **0.82** | *0.96* | *0.96* | **0.97** | **0.92** | **0.92** | **0.92** | **0.96** | *0.94* | 0.95 |
| Tiffauges | *0.80* | **0.81** | **0.81** | *0.91* | **0.94** | 0.92 | **0.96** | **0.96** | **0.96** | **0.96** | *0.91* | **0.96** |
| Cholet | **0.78** | 0.70 | *0.67* | **0.92** | *0.76* | 0.78 | *0.80* | 0.84 | **0.87** | **0.89** | *0.56* | 0.78 |
| Roussay | **0.82** | 0.80 | *0.76* | **0.92** | 0.84 | 0.84 | *0.92* | 0.92 | **0.93** | **0.94** | *0.71* | 0.87 |
| St-Crespin-sur-Moine | **0.83** | **0.83** | *0.79* | **0.92** | 0.86 | 0.86 | **0.95** | *0.94* | **0.95** | **0.94** | *0.75* | 0.90 |
| Clisson | *0.86* | **0.87** | 0.86 | 0.95 | **0.96** | 0.95 | **0.96** | **0.96** | **0.96** | **0.97** | **0.97** | **0.97** |
| Tillières | 0.66 | **0.67** | 0.66 | *0.87* | **0.90** | 0.89 | **0.86** | **0.86** | **0.86** | *0.84* | 0.90 | **0.92** |
| St-Fulgent | **0.89** | **0.89** | **0.89** | 0.99 | *0.98* | **1.00** | **0.94** | **0.94** | **0.94** | *0.94* | **0.95** | **0.95** |
| St-Georges-de-Montaigu | **0.71** | **0.71** | **0.71** | *0.94* | 0.95 | **0.97** | **0.87** | **0.87** | **0.87** | *0.89* | **0.92** | 0.90 |
| Remouillé | *0.76* | **0.81** | 0.79 | *0.94* | **1.00** | 0.98 | *0.92* | **0.93** | *0.92* | **0.94** | **0.94** | **0.94** |
| Nantes | *0.79* | **0.81** | 0.80 | *0.94* | **0.95** | *0.94* | **0.94** | **0.94** | **0.94** | **0.97** | **0.97** | *0.96* |







**Figure 4: Interannual streamflow regime over 2008-2020 for the 13 gauge stations, observed and simulated by the three model**
**versions using SAFRAN as input**




### 3.2 Evolution of uninfluenced streamflow under climate change

In this section, hydrological projections are presented independently of water uses within the catchment, i.e. in its "Uninf" configuration. This means that the hydrological model, as calibrated earlier (see "Calib" model), is applied within the context

of climate change, but without considering water uses, either during the historical or the future periods. This approach enables the assessment of the natural streamflow evolution in response to climate change. The results are presented at the Management Unit outlets to facilitate the interpretation of subsequent water uses results, which are aggregated at these scales. This is also justified by the fact that the water management is done as the MU scale.

Figure 5 illustrates the interannual streamflow regime, uninfluenced by water use, for five climate projections, three future

periods, and the reference period, at the Sèvre Nantaise catchment outlet in Nantes. It is evident from the Figure 5 that the streamflow evolution is contingent on both the climate projections and the time period. Overall, the evolution appears limited for projection A1, which represents the scenario with the lightest climate evolution. Projection F4 shows an increase in streamflow during winter for all periods, as well as an increase at the start of spring for the 2016-2045 period. Evolutions for the other projections are more contrasted, in accordance with the climate evolutions described in Table 3. Generally,

streamflow tends to increase in winter and early spring (especially for projections C1 and C2 for all time periods, except 2036-2065, which show the highest increases in winter precipitation), and to decrease in autumn. However, no clear trend emerges when comparing the different future periods. These trends are consistent across the rest of the catchment, as seen in Appendix D for the Cholet and St-Fulgent stations. Low-flow evolutions are not visible in Figure 5 or Appendix D due to the already low streamflow values.

Figure 6 shows that the evolution of several indicators is highly dependent on the climate projection for the 2056-2085 period, which we focus on as it is particularly relevant for water managers facing significant climate changes. Regarding average streamflow (QA), the A1 and C1 projections - which indicate stagnation and increased annual precipitation, respectively - lead to slight to moderate increases in this indicator for most Management Units. In contrast, other projections lead to limited evolutions or even decreases. For high flows (Q90), projections A1, C1 and C2 show an increase, while B3 and F4 indicate a

decrease. The projections for low flows (QMNA5 and Q05) and mid-flows (Q50) show more intense evolutions. It is clear that all projections lead to a decrease for these three indicators, consistent with the decrease in summer precipitation. Only C1, which predicts an increase in annual precipitation and a stagnation in summer precipitation, leads to an increase in one of these indicators, Q50. This implies that for low flows, even a favourable climate evolution would result in decreased water resources, especially during dry periods. Regarding spatial patterns, only MU3 and MU4, located along the Moine River, show a distinct

pattern, with less intense evolutions, except for mid flows.



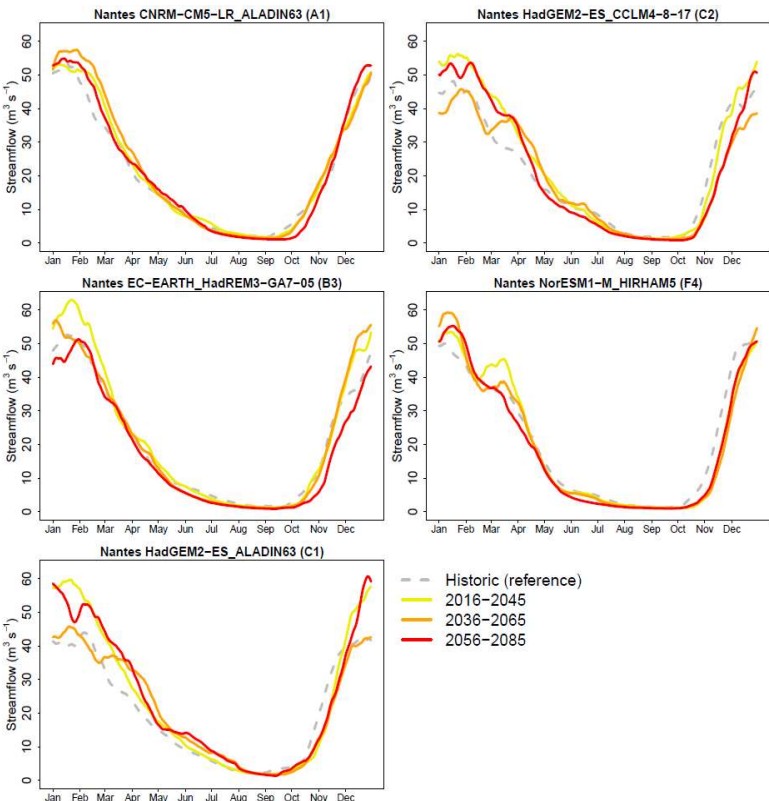

**Figure 5: Interannual streamflow regime for the projected uninfluenced streamflow for the Nantes gauge station (main outlet of the catchment) for three future periods for the five climate projections.**





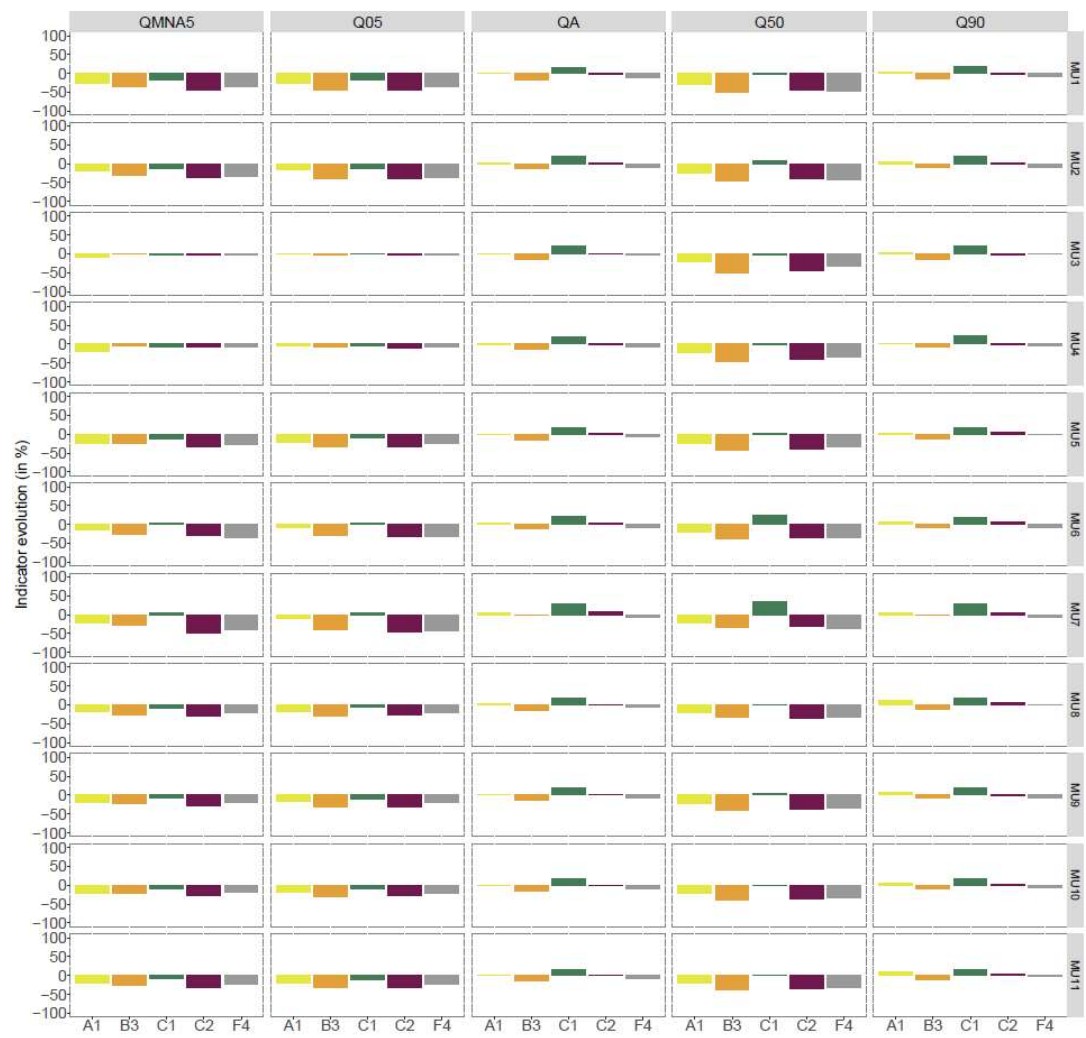

**Figure 6: Evolution of the uninfluenced hydrological indicators (in columns) under five climate projections for the 11 Management Units (in rows) of the Sèvre Nantaise for 2056-2085 compared to 1976-2005.**



**3.3 Evolution of water demand under climate change and water demand scenarios for 2056-2085**

Here, we present the evolution of water demand under climate change, along with three water demand scenarios (Figure 7). It is important to remind that water demand refers to the theoretical amount of water requested by the different uses, independent of actual water availability or climate conditions. As shown in Figure 7, only the demand for irrigation water is impacted by the choice of the climate projection, as it is the only water use for which water demand vary regarding climate projection. This makes sense, as none of the other water uses are strongly influenced by climate, except for cattle watering that increases a bit when air temperature exceeds 30 °C. Indeed, irrigation tends to increase more under climate projections where precipitation decreases the most (B3, C2 and F4). We can also observe that irrigation and cattle watering are the water uses exhibiting the largest relative evolutions. However, it is important to remember that the volumes of water concerned by these changes are not of the same order (see Table 4 for observed volumes in 2008-2020), with irrigation and drinking water accounting for the overall largest withdrawn volumes. When comparing water demand scenarios, we find that the alternative scenario leads to overall decreases or stagnation of water demand, while the trend scenario leads to an overall increase, except for cattle watering, which is already declining. Differences between MUs mainly relate the fact that specific water uses are absent, such as drinking water for many MUs, or industry for MU10 and MU11, a result from the multiplying factors applied to translate the water use scenarios into actual demands (Appendix C).



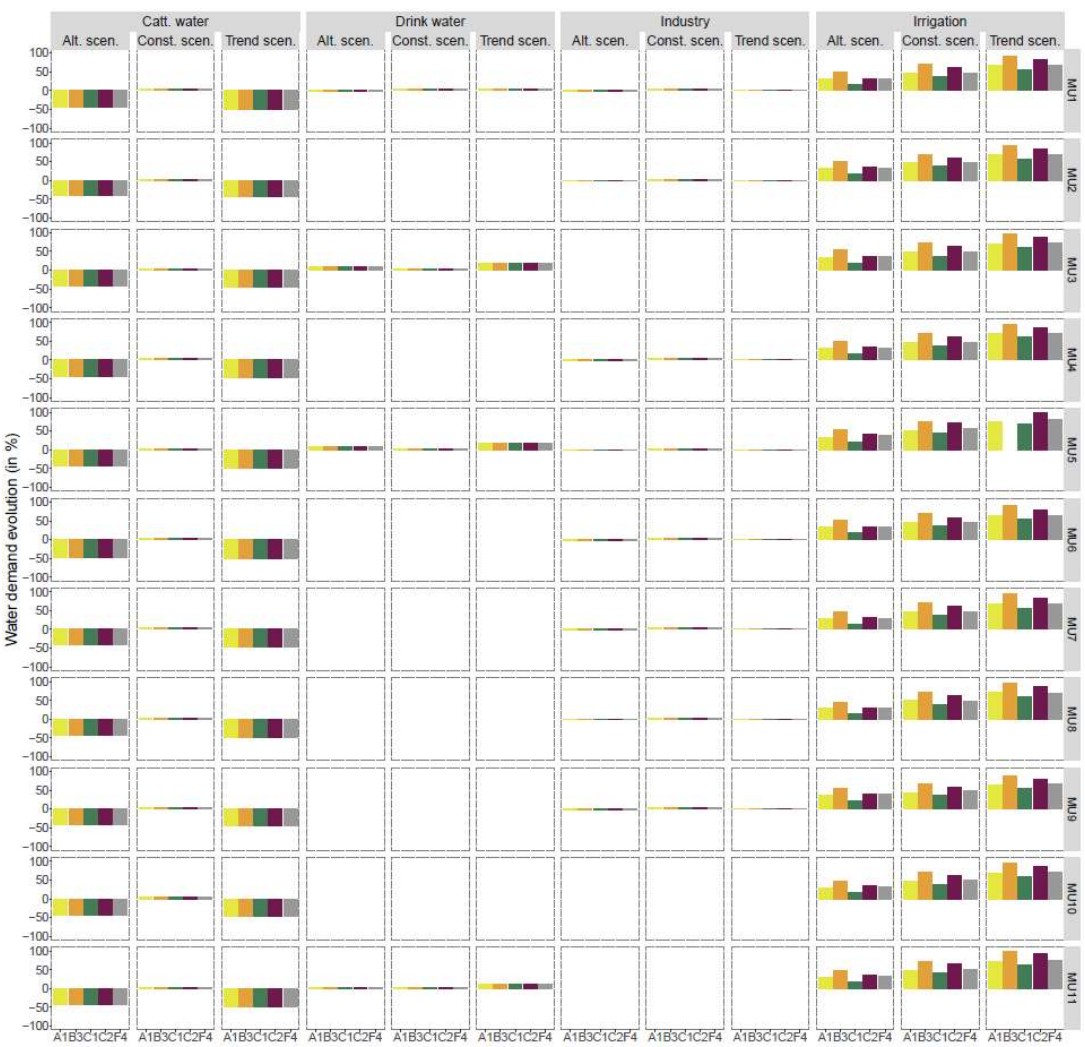

**Figure 7: Relative evolution of water demand for the different uses and scenarios (in columns) under five climate projections for the 11 management units (in rows) of the Sèvre Nantaise for 2056-2085 compared to 1976-2005. Empty panels represent the cases where a water use is not present in a MU.**





**3.4 Evolution of streamflow under climate change and water demand scenarios around 2050**

In this section, we present the evolution of streamflow both under climate change and three water demand scenarios. We begin by comparing the streamflow evolution under climate change for three water use scenarios with that under uninfluenced conditions (Figure 8). Figure 8 clearly shows that, regardless the MU, average streamflow (QA) is primarily impacted by climate change, with only minor influence from the water demand scenarios or the consideration of water use in general. This

means that water consumption is not large enough to alter the total amount of water flowing into the rivers at the MU scale. This may be explained by the fact that, except for irrigation, a significant part of withdrawn water is released in the same MU. Nevertheless, the situation is different for the QMNA5 low-flow indicator. For QMNA5, while climate change remains a major driver of the streamflow indicators evolution, water demand scenarios also influence its evolution. For most MUs – except for MU2, MU5 and MU7 – the alternative scenario results in the highest increases or to the smallest decreases in QMNA5. It is

expected, as the alternative scenario involves the greater efforts to decrease water withdrawals in the Sèvre Nantaise catchment, as shown in the water demands evolution (Figure 7). In case where the alternative scenario is the most beneficial, the trend scenario is the least favourable (except for MU6), as it leads to an overall increase in water demand for drinking water and irrigation, and a decrease in demand from industry and cattle watering, both of which require less water. The specific case of MU6 is due to the large amount of water used for cattle watering relative to other uses in this MU, which is expected to reduce

by more than 30 % in the trend scenario. In MU2, the constant scenario has less negative impact on QMNA5 compared to the other scenarios. In this MU, irrigation is the main withdrawal, while wastewater treatment plants are the main source of water releases. This suggests that, under the alternative scenario, the least increase of irrigation may be offset by a reduction in volumes from wastewater treatment plants, due to lower drinking water supply from other catchments.







**Figure 8: Evolution of the QA (columns 1 and 3) and QMNA5 (columns 2 and 4) indicators for the 11 Management Units (rows) for the influenced simulations (y axis) against the uninfluenced simulations (x axis). The scenarios are depicted with different symbols and the climate projections with different colours. Since no scenario is accounted for the uninfluenced indicator, the values are duplicated for each influenced indicator. The evolutions are calculated from 2056-2085 to 1976-2005.**






### 3.5 Evolution of water demand satisfaction for 2056-2085

Figure 9 presents the water demand satisfaction levels for each water use, each MU, each climate projection, and each water demand scenario. The analysis reveals that water uses are not uniformly affected by climate change and water demand scenarios. For instance, drinking water satisfaction appears to be barely affected by both factors. This can be attributed to the

prioritization of drinking water supply. Conversely, irrigation – the lowest-priority water use – is significantly impacted by climate change.  Compared to the reference period (illustrated in grey), the irrigation satisfaction rate declines under all climate projections except C1 (green), where the decrease is relatively modest and can even lead to an increase in some MUs. The alternative scenario results in higher irrigation satisfaction rates than the other scenarios. In between, industry and cattle watering, which have an intermediate priority among water uses, are only slightly affected by climate change, with the

exception of MU9 for cattle watering. The alternative scenario provides only limited benefits in these sectors.

Irrigation largely depends on water use from small reservoirs, whose filling levels are also affected by climate change and water use scenarios (Figure 10). While the C1 climate projection results in a modest decline in the filling rate of these dams, due to favourable precipitation conditions, all other projections leads to substantial reductions. These declines can be partially, though not entirely, mitigated by the alternative scenario. The MUs unaffected by these decreases are rare (e.g., MU8).



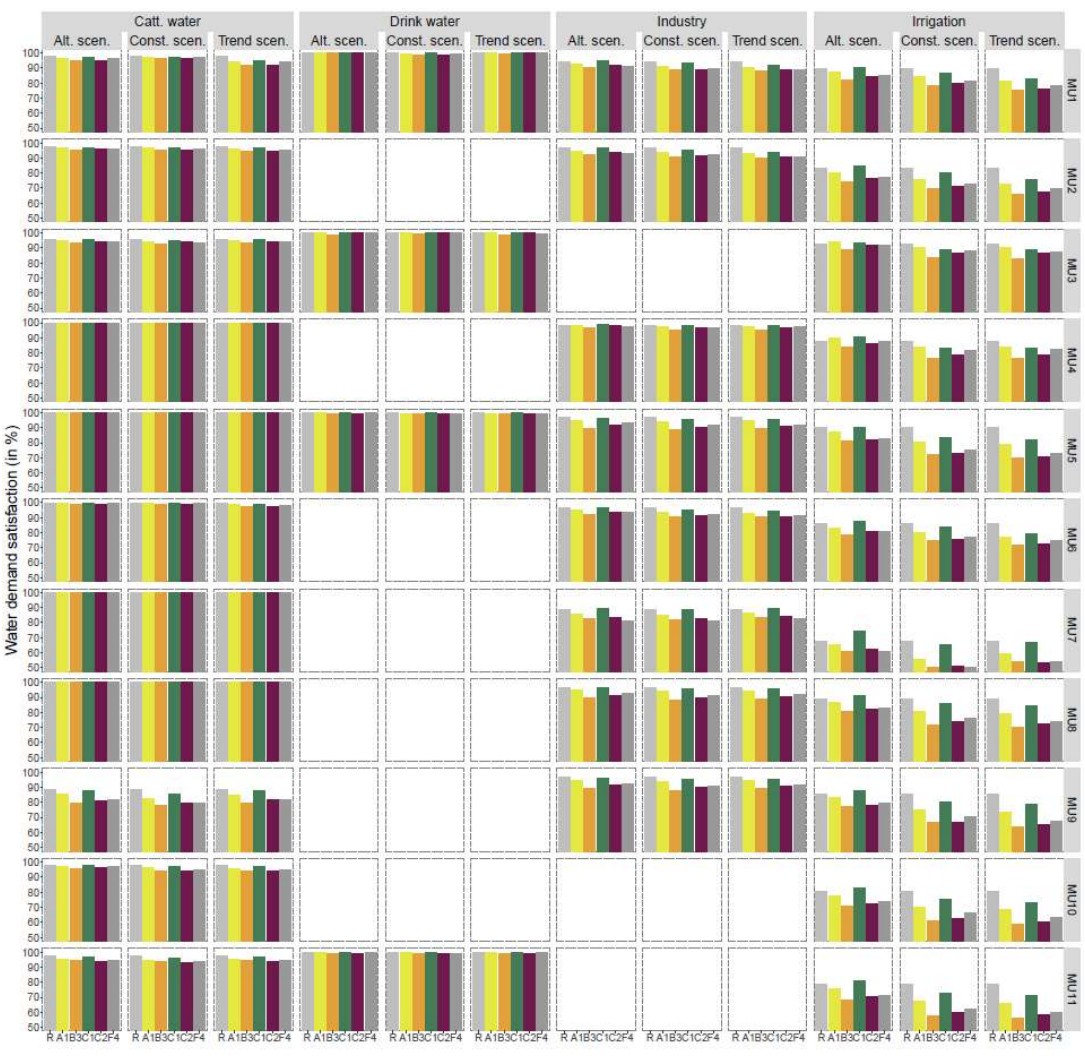

**Figure 9: Water demand satisfaction (in %) for the different uses and scenarios (in columns) under five climate projections for the 11 MUs (in rows) for 2056-2085 (projections A1, B3, C1, C2, F4) and for 1976-2005 (the reference historical period, R). Empty panels represent the cases where a water use is not present in a MU. The scale used for satisfaction goes from 50 to 100 %.**






**Figure 10: Probability of filling small irrigation dams for the three scenarios under five climate projections for the 11 MUs for 2056-2085 (under the three scenarios) and 1976-2005 (historical).**



## 470    4    Discussion

### 4.1    Consequences of climate change and water use scenarios for water resources management on the Sèvre Nantaise catchment

The future evolution of water resources in the Sèvre Nantaise catchment appears to be contingent on climate-driven changes. Precipitation is expected to increase during winter and decrease during summer, impacting streamflow patterns similarly.
However, the magnitude of these changes varies across different climate projections. Four out of five projections indicate a substantial decline in low-flow indicators, while the C1 projection suggests a more limited impact. Regarding total water resources (mean streamflow) or high flows, only the C1 projection results in an increase, while the others predict a decrease, albeit less pronounced than the evolution of low flows. These results are coherent with the results of the recent nation-wide Explore2 project (Sauquet et al., 2025). Water use scenarios illustrate a general reduction in water demand, except for
irrigation. This decline is attributable to either enhanced network efficiency or a reduction in cattle populations, particularly in case additional measures are implemented (alternative scenario). However, irrigation demands increase in all scenarios, albeit to a limited extent in the alternative scenario, which concords with Konzmann et al. (2012) or Aslam et al. (2025). This rise is attributable to higher crop evapotranspiration due to increasing air temperatures and declining soil moisture, resulting from reduced precipitation. Once again, only the C1 projection limits this increase.

The findings pertaining to average streamflow are noteworthy: its evolution is driven by climate change rather than water withdrawals, which is consistent with Lemaitre-Basset et al. (2024), for instance. While some water withdrawals may be substantial, they constitute a negligible proportion of the annual water resource. Furthermore, these withdrawals are partially compensated by water releases within the same Management Units. Conversely, both water withdrawals and climate change have a major impact on the low-flow evolution. The evolution is complex, with some MUs experiencing a mitigated impact
of climate change due to water management, while others are less impacted. The alternative scenario generally leads to less negative evolutions, but not always. Among all climate projection, the C1 is the least detrimental. However, these evolutions in water resources and demand lead to overall decrease of water demand satisfaction throughout the Sèvre Nantaise catchment. Although the primary intention of the alternative scenario was to propose adaptation strategies to help compensate for the impact of climate change, it is possible that the discussions with stakeholders may have limited this objective. The adaptation
scenario may have been toned down due to the reluctance of certain actors to disadvantage their water demand sector, or perhaps due to the challenges associated with considering realistic, drastic changes in water usage. Consequently, this has led to a limitation in the contrasts between the results of the different water use scenarios, thereby hindering the identification of a "desirable future". While the priority water use, i.e. drinking water, remains largely safeguarded, lower-priority uses, particularly irrigation, face a drastic reduction in satisfaction. The most favourable combination of climate projection and water
use scenario (C1 projection under the alternative scenario) still leads to a limited deterioration of the situation. In addition, it





is highly improbable that small reservoirs are likely to maintain current filling rates in the future, even under the most favourable conditions. These facts are concerning, as the most optimistic scenario requires both strong and proactive local action to curtail the escalating water demand and a global reaction to climate change, characterised by limited climate change response to greenhouse gas concentrations. In any other case, restrictions on water use, impacting sectors such as irrigation,

industry, and cattle watering, will become severe, resulting in significant local economic consequences. It can be posited that local stakeholders possess the capacity to mitigate the repercussions of climate change; however, the efficacy of existing solutions may be insufficient. In addition, the consequences for aquatic life and more generally environmental issues might be severe.

### 4.2 On the complexity of a real-life catchment-scale water resources management

The in-depth representation and investigation of a real-life catchment-scale water resources management system is a subject of great interest. Indeed, such systems are characterised by multi-factorial and complex ensembles of careful decisions and interactions between water uses and the evolution of water resources. Catchment-scale integrated water resources modelling is therefore the only means to accurately describe these intricate relationships. Larger and coarser representations of integrated water resources management can offer valuable insights. They are useful for providing general depictions of a situation and

identifying key drivers in the context of evolving climate and water uses. For example, they are applied in the development of country-scale adaptation plans. However, these models fall short in accurately representing local management practices and in ensuring the acceptance of their conclusions by local stakeholders. Furthermore, adaptation strategies are predominantly driven by the specific characteristics of the local context, in addition to national policy directives.

Deploying a detailed modelling tool remains challenging, as it requires a comprehensive understanding of the catchment

functioning. First, the access to water use data is often difficult, with limited available databases typically being incomplete or not available in digital format, covering short time periods or featuring inadequate temporal resolution. This can result in the formulation of assumptions to represent water uses within a catchment, or even hinder the evaluation of the modelling. Second, human decisions-making has a strong impact on water resources. Farmers' decisions regarding irrigation, as well as authorities' decisions to restrict water withdrawals and users' compliance, are key factors. While agent-based models (van Oel et al., 2010;

Sousa et al., 2025) incorporate these behaviours, doing so necessitates a high level of model intricacy. Furthermore, it necessitates in-depth in situ investigation to comprehend human behaviours, which is difficult to undertake in large-scale basins.

Another key consideration is the vast amount of data generated by water resources management models. At each time step (e.g., day), for each climate projection, each water use scenario, each category of water use, and each MU, a wide range of

water fluxes (both natural and related to water uses) and decisions (e.g. irrigation triggering or water uses restriction) are simulated. This results in an extensive dataset with billions of values and numerous potential interactions between them, which are intricate to depict.





### 4.3 About uncertainties

Uncertainty quantification is a critical issue when dealing with climate change impacts and water demand scenarios. In this study, we did not perform a systematic quantification of the contribution of the different modelling steps to the total uncertainty, as could be done with an analysis of variance (ANOVA) method (see e.g. Evin et al., 2025). While the application of such a method is increasingly prevalent in studies that integrate climate projections and hydrological modelling (see, for example, Lemaitre-Basset et al., 2024), its use in conjunction with water use modelling within the modelling chain, as in the present study, remains relatively uncommon. This may warrant further investigation. Nevertheless, the present study helps identifying some of the primary drivers of uncertainty in water resources, water demand, influenced streamflow, and water demand satisfaction evolutions, as discussed in the previous sections. Specifically, climate projections appear to be the dominant source of uncertainty for natural water resources, influenced streamflow, water use satisfaction, and irrigation water demand evolutions. Water use scenarios introduce a certain degree of uncertainty regarding water demand, influenced streamflow, and water use satisfaction evolutions, albeit generally at a lower level than climate projections. This underscores the need for further research in both climate and water use modelling. It should be noted that the uncertainty related to greenhouse gas emission scenarios and hydrological modelling was not assessed in this study for the sake of simplicity. However, previous studies have shown that these modelling steps can constitute a substantial source of uncertainty, especially for low flows (Vidal et al., 2016).

The substantial uncertainties identified in such studies can be perplexing for stakeholders. Indeed, the sometimes-opposite trends in water resources evolution, or the limited effect of adaptation strategies on the actual evolution of influenced water resources or water demand satisfaction, may lead stakeholders to believe that the situation is too uncertain to take action. While the primary focus of this study was on the 2056 – 2085 period, it was observed that the evolution of uninfluenced water resources in the XXI$^{st}$ century may exhibit varying patterns prior to this period, with potential more positive phases as well as potentially more negative phases. These fluctuations are attributed to climate variability, rather than variability in water use, which was assumed to evolve progressively.

### 5 Conclusion

The present study has sought to assess the impact of climate change and water demand evolution on the Sèvre Nantaise catchment, a pluvial temperate watershed in western France. With a set of national state-of-the-art contrasted climate projections, it reveals that while the future evolution of annual natural streamflow remains uncertain, a decrease of low flows is highly likely. A comprehensive information and data collection process on water uses within the catchment led to the formulation of coherent water demand scenarios. An integrated water resources model was then established to investigate the combined impact of climate change and water use evolution on streamflow and water demand satisfaction. The analysis yielded several key findings. Both influenced low flows and water demand satisfaction are expected to deteriorate. The impact of water demand evolution, through an alternative scenario leading to water demand reduction, was deemed to be limited on streamflow



and water demand satisfaction evolution, compared to the impact of climate evolution. It was also observed that irrigation will be the most impacted water use, due to its low priority.

Such a catchment-scale integrated water resources modelling effort provides a more detailed understanding of the local drivers of climate change impact and adaptation than efforts at a larger scale, which offer a broader but less detailed picture. Notwithstanding, Sèvre Nantaise water managers are currently developing a catchment water management initiative (in

French: projets de territoire pour la gestion de l'eau, PTGE), despite the uncertainties identified in this work. This process is law-based and predicated on a comprehensive, joint approach to water resources for every water use within a coherent area from a hydrological or hydrogeological point of view. The resultant commitment on the part of all users in the area (drinking water, agriculture, industry, inland navigation, energy, fisheries, recreational uses, etc.) is to achieve a long-term balance between needs and available resources. This must be achieved while respecting the proper functioning of aquatic ecosystems

and anticipating and adapting to climate change.

**Author contribution**

LS, AT, LM and GT conceptualized the study. LS, AT, GTa and GTh collected and curated the data. LS, AT, GTa and GTh performed most analyses and all other authors contributed to additional analyses. AT, LM and GTh acquired the funding. LS, AT, LM, GTa and GTh designed the methodology and prepared the original draft. All authors reviewed and edited the

manuscript.

**Competing interest**

The authors declare that they have no conflict of interest.

**Acknowledgments**

All local stakeholders that share either water uses data or participated to the workshops aiming at defining the water demand

scenarios are deeply thanked for their contribution. We also thank the HydroPortail for providing hydrological data and Météo-France for providing both the atmospheric reanalysis data and the climate projection data. Charles Perrin, David Dorchies and Evelyne Talès (INRAE) are thanked for their participation on other aspects of the study that were not presented in this article. Finally, with thank the Loire-Bretagne Water Agency and the EPTB Sèvre Nantaise for partially funding this work.

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

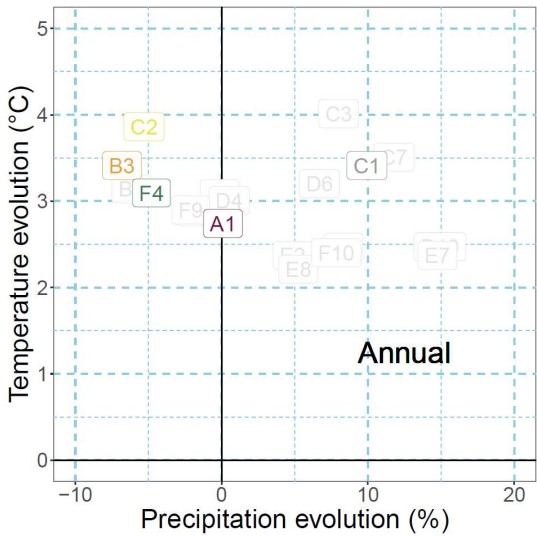

**Figure A1: Evolution of the annual average precipitation (P) and temperature (T) for RCP 8.5 and for 2056-2085 relatively to 1976-2005. The colours highlight the five selected projections and allow to compare them to the complete Explore2 dataset (not detailed here). The letters stand for the global climate models and the numbers for the regional climate models.**



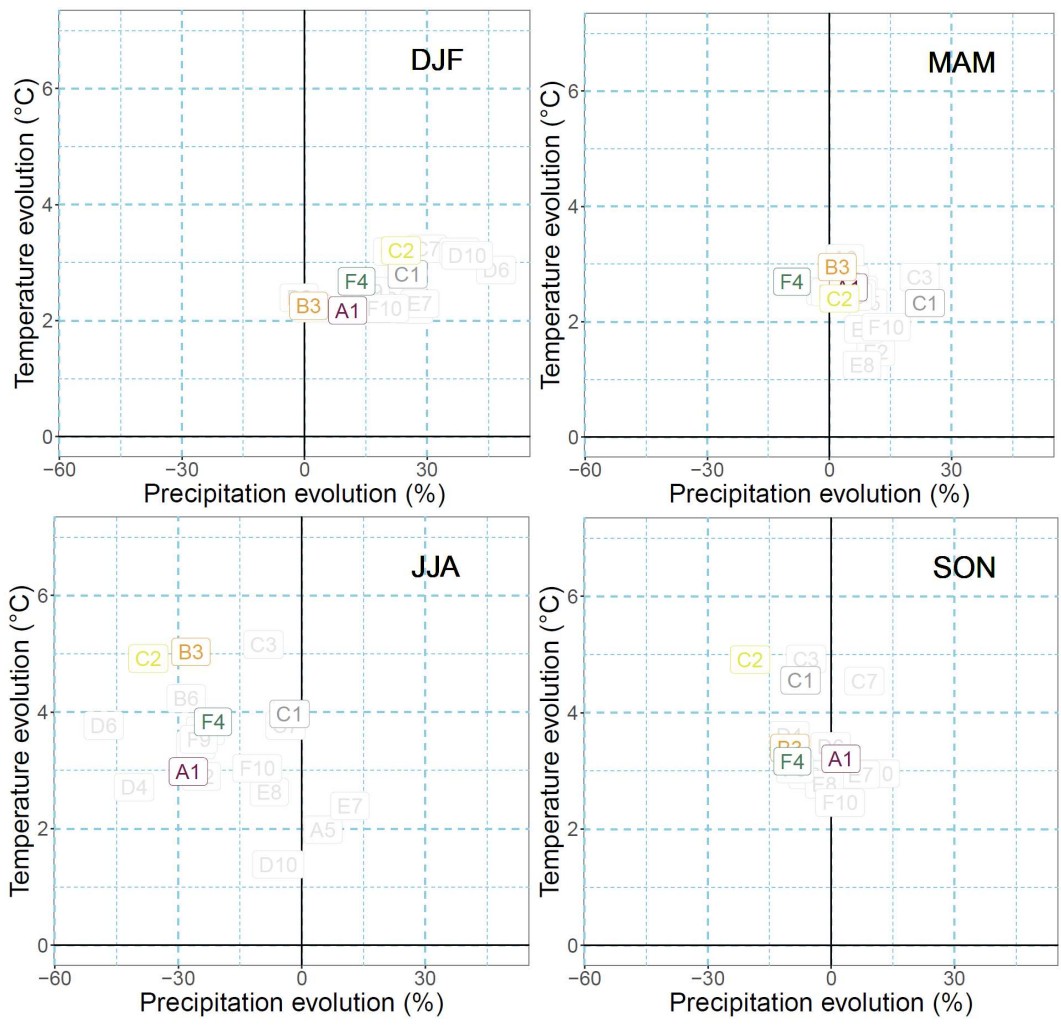


**Figure A2: Evolution of the seasonal average precipitation (P) and temperature (T) for RCP 8.5 and for 2056-2085 relatively to 1976-2005. The colours highlight the five selected projections and allow to compare them to the complete Explore2 dataset (not detailed here). DJF: December-January-February; MAM: March-April-May; JJA: June-July-August; SON: September-October-November.**




## Appendix B. Water demand and release models

**Table B1: Water demand models and variables.**

| Water demand sector | Equation for location $a$ and day $d$. Areas can be either municipalities or plots (for irrigation). The water demands are thereafter summed up over the diverse areas a of each hydrological modelling unit | Variables |
|---|---|---|
| Cattle watering | The daily water demand made on drinking water for cattle for location a and day d is: $$Q_{CW,DW}(a,d) = \sum_{i=cattletype} n(i,a,y) * Dem(i,y) * R_{DW}(i,y) * R_D(d)$$ The daily water demand made on the natural environment for cattle for location a and day d is: $$Q_{CW,NE}(a,d) = \sum_{i=cattletyp} n(i,a,y) * Dem(i,y) * (1 - R_{DW}(i,y)) \\ * R_D(d)$$ | $n(i,a,y)$ is the number of heads for cattle type $i$, location $a$ and year $y$ [-] $Dem(i,y)$ is the demand per head for cattle type $i$ and year $y$ [m³ y⁻¹] $R_{DW}(i,y)$ is the fraction of water demand made on drinking water for cattle type $i$ and year $y$ [-] $R_D(d)$ is the fraction of yearly water demand made on day $d$ [-] |
| Industry | The daily water demand made on drinking water for industry for location a and day d is: $$Q_{Ind,DW}(a,d) = \sum_{i=industr\ (a)} Dem_{Ind}(i,y) * R_{DW2}(i,y) * R_{D2}(d)$$ The daily water demand made on the natural environment for industry for location a and day d is: $$Q_{Ind,NE}(a,d) = \sum_{i=industry(a)} Dem_{Ind}(i,y) * (1 - R_{DW2}(i,y)) * R_{D2}(d)$$ | $Dem(i,y)$ is the demand for industry $i$ and year $y$ [m³ y⁻¹] $R_{DW2}(i,y)$ is the fraction of water demand made on drinking water for industry $i$ and year $y$ [-] $R_{D2}(d)$ is the fraction of yearly water demand made on day $d$ [-] |



| Irrigation | The daily water demand made for irrigation for location a and day d is: $$Q_{Irrig}(a,d) = \sum_{i=croptyp} Surf(i,a,y) * Dem_{unit}(i,d) * R_{Irrig}(i,y)$$ | $Surf(i,a,y)$ is the cultivated surface for crop $i$, location $a$ and year $y$ [-] $Dem_{unit}(i,d)$ is the demand for 1 m² for crop $i$ on day $d$ of year $y$ [m³ d⁻¹] $R_{Irrig}(i,y)$ is the fraction of irrigated surface for crop $i$ and year $y$ [-] |
|---|---|---|
| Vine spraying | The daily water demand made for vine water spraying for location a and day d is: $$Q_{vine}(a,d) = Surf(vine,a,y) * Dem_{unit}(vine,d) * R_{sprayed_{vine}}(i,y)$$ | $Surf(vine,a,y)$ is the cultivated surface for vine, location $a$ and year $y$ [m²] $Dem_{unit}(vine,d)$ is the demand vine spray under freezing conditions for 1 m² and day $d$ [m³ d⁻¹m⁻²] $R_{sprayed,vine}(i,y)$ is the fraction of vine surface concerned by spraying in freezing condition and year $y$ [-] |
| Drinking water | The daily water demand made for drinking water for location a and day d is: $$Q_{DW}(a,d) = Q_{CW,DW}(a,d) + Q_{ind,DW}(a,d) + pop(a,y) * Cons_{unit}$$ The daily water demand is then aggregated for each withdrawal point proportionally to its origin. | $pop(a,y)$ is the population of location $a$ and year $y$ [inhab.] $Cons_{unit}$ is the daily unit consumption per human being [m³ d⁻¹ inhab.⁻¹] |




Table B2: Water release models and variables

| Water release sector | Equation for location a and day d (the water releases are thereafter summed up over the diverse areas a of each hydrological modelling unit) | Variables |
|---|---|---|
| Industry | The daily water release made in the drinking water network for industry for location a and day d is: $$Rel_{Ind,DW}(a,d) = \sum_{i=industry(a)} Rel_{Ind}(i,y) * R_{DW3}(i,y) * R_{D2}(d)$$ The daily water release made in the drinking water network for industry for location a and day d is: $$Rel_{Ind,NE}(a,d) = \sum_{i=industry(a)} Rel_{Ind}(i,y) * (1 - R_{DW3}(i,y)) * R_{D2}(d)$$ | $Rel_{Ind}(i,y)$ is the water release for industry $i$ and year $y$ [-] $R_{DW3}(i,y)$ is the fraction of water release made in the drinking water network for industry $i$ and year $y$ [-] $R_{D2}(d)$ is the fraction of yearly water release made on day $d$ [-] |
| Wastewater treatment | The daily water treatment release for location a and day d is: $$Rel_{WW}(a,d) = Rel_{Ind,DW}(a,d) + Q_{DW}(a,d) + P(P > 10mm)$$ The daily water treatment release is then aggregated for each sewage plant proportionally to its destination. | $P(P > 10mm)$ is the the daily rainfall higher than 10 mm [m³ d⁻¹] |
| Drinking water network losses | The daily drinking water network losses for location a and day d are: $$Rel_{DWloss}(a,d) = Q_{DW}(a,d) * Eff(a,y)$$ | $Eff(a,y)$ is the drinking water network efficiency for location $a$ and year $y$ [-] |

Table B3: Dams management rules.

| Dam | Minimum outflow | Condition for minimum outflow | Other management rules |
|---|---|---|---|
| Ribou-Verdon | 200 l s⁻¹ | From September to May | Objective to maintain a stock of 3,75 Mm³ out of 5 Mm³ from December to February to mitigate floods |
| | 400 l s⁻¹ | From June to August | |
| | 100 l s⁻¹ | If stored water is lower than 5 Mm³ | |
| Bultière | 160 l s⁻¹ | All year round until 2020 | - |
| | 260 l s⁻¹ | From November to March past 2020 | |
| | 160 l s⁻¹ | In April and May past 2020 | |
| | 100 l s⁻¹ | From July to October past 2020 | |






**Appendix C.    Water demand and release parameters for each scenario**

Constant scenario

All water demands and releases remain constant compared to 2020.




**Table C1: Trend scenario**

| Item | Spatial scale | Unit | Reference | 2016-2045 | 2036-2065 | 2056-2085 |
|---|---|---|---|---|---|---|
| \multicolumn{7}{Drinking water and sewage treatment} |||||||
| Population | Heterogeneous over the catchment | Percentage of evolution | 2020 | From 0 to +0.5 % between 2023 and 2059 following a linear trend. Stable after 2059. Main cities follow a higher increase (+0.1 %), while rural areas follow a lower increase (-0.1 %) | | |
| Consumption per unit | | | | Stable overall, but +50 L/inhabitant/day for rural areas, -50 L/inhabitant/day for urban areas | | |
| Drinking water network efficiency | Whole catchment | | | +0.013 % per year | | |
| Partition between collective and individual sewage treatment | | Rate | | Incoming population considered as within the collective network | | |
| Inter-catchment transfers | | - | | Unchanged except if already planned | | |
| \multicolumn{7}{Agriculture} |||||||
| Dairy cows | Whole catchment | Percentage of evolution of the number of heads | 2020 | -0.6 % per year | | |
| Suckler cows | | | | -2.0 % per year | | |
| Calves | | | | -0.7 % per year | | |
| Porks | | | | -0.4 % per year | | |
| Poultry | | | | -2.0 % per year | | |
| Consumption per unit | | L d$^{-1}$ head$^{-1}$ | | Unchanged | | |
| Part of water withdrawal coming from the drinking water network | | Rate | | Unchanged | | |
| Total cultivated area | | Percentage of evolution | 2020 | -0.2 % per year | | |
| Crop rotation | | Type of crop | | 10 % reduction over 10 years in vineyards and forage crops, replaced by wheat, maize, rapeseed and market gardening (in equal proportions, depending on what was already present) | | |
| Vine water spraying | | - | | On half of vineyards | | |
| Irrigation practices | | - | | Unchanged | | |



| Irrigated surfaces | | km² | | +5 % | +10 % | +15 % |
|---|---|---|---|---|---|---|
| Industries | | | | | | |
| Industrial activity | Whole | Percentage | 2020 | +1 % | +2 % | +3 % |
| Consumption per unit | catchment | of evolution | | -2 % | -4 % | -6 % |

8000




**Table C2: Alternative scenario**

| Item | Spatial scale | Unit | Reference | 2016-2045 | 2036-2065 | 2056-2085 |
|---|---|---|---|---|---|---|
| Drinking water and sewage treatment | | | | | | |
| Population | Heterogeneous over the catchment | Percentage of evolution | 2020 | From 0 to +0.5 % between 2023 and 2059 following a linear trend. Stable after 2059. Main cities follow a higher increase (+0.2 %), while rural areas follow a lower increase (-0.2 %) | | |
| Consumption per unit | | | | Stable overall, then -1 % per year from 2040 to 2050 | | |
| Drinking water network efficiency | Whole catchment | | | +0.08 % per year | | |
| Partition between collective and individual sewage treatment | | Rate | | Incoming population considered as within the collective network | | |
| Inter-catchment transfers | | - | | Unchanged except if already planned | | |
| Agriculture | | | | | | |
| Dairy cows | Whole catchment | Percentage of evolution of the number of heads | 2020 | -0.6 % per year | | |
| Suckler cows | | | | -1.4 % per year | | |
| Calves | | | | -0.7 % per year | | |
| Porks | | | | -0.5 % per year | | |
| Poultry | | | | -1.4 % per year | | |
| Consumption per unit | | L d⁻¹ head⁻¹ | | Unchanged (except during days with air temperature > 30 °C) | | |
| Part of water withdrawal coming from the drinking water network | | Rate | | +5 % | +10 % | +20 % |
| Total cultivated area | | Percentage of evolution | 2020 | Unchanged | | |
| Crop rotation | | Type of crop | | Wheat is partly replaced by barley, and maize by sorghum, at a rate of 0.5 % per year. | | |
| Vine water spraying | | - | | On half of vineyards | | |
| Irrigation practices | | - | | Unchanged | | |
| Irrigated surfaces | | km² | | +5 % | +10 % | +15 % |
| Industries | | | | | | |



| Industrial activity | Whole | Percentage | 2020 | +1 % | +2 % | +3 % |
|---|---|---|---|---|---|---|
| Consumption per unit | catchment | of evolution | | -4 % | -6 % | -8 % |

805



## Appendix D.    Uninfluenced future streamflow regimes for additional stations

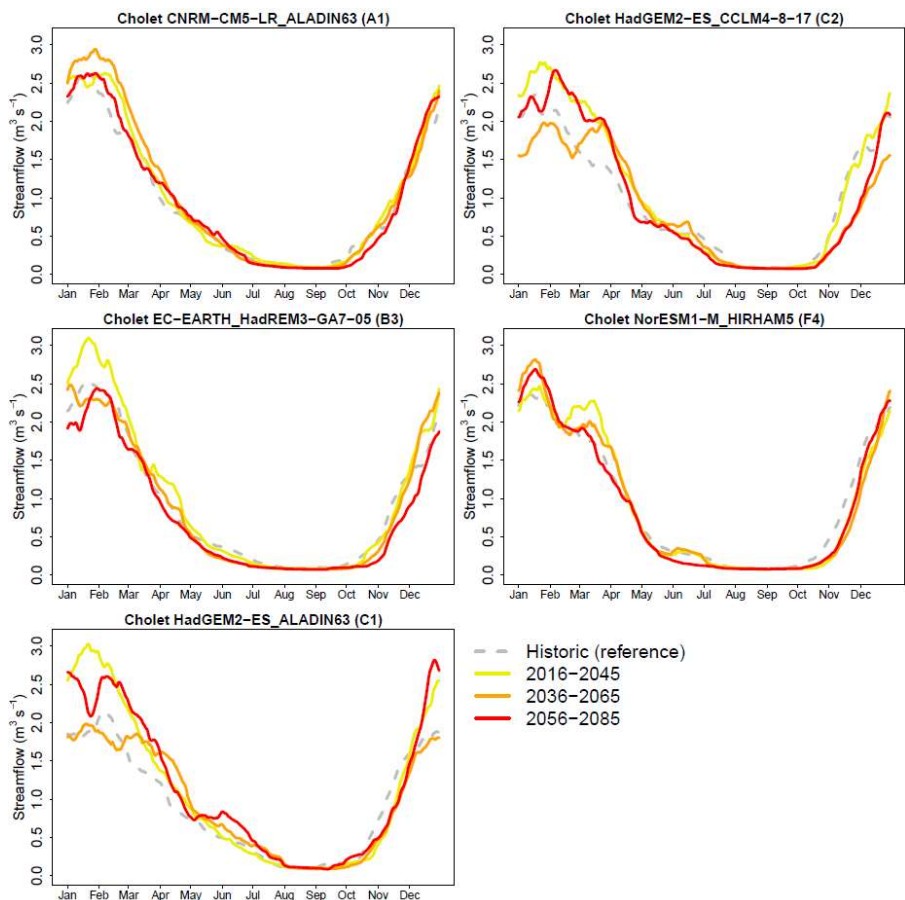

**Figure D1: Interannual streamflow regime for the projected uninfluenced streamflow for the Cholet gauge station (downstream of the Ribou-Verdon dams) for three future periods for the five climate projections.**





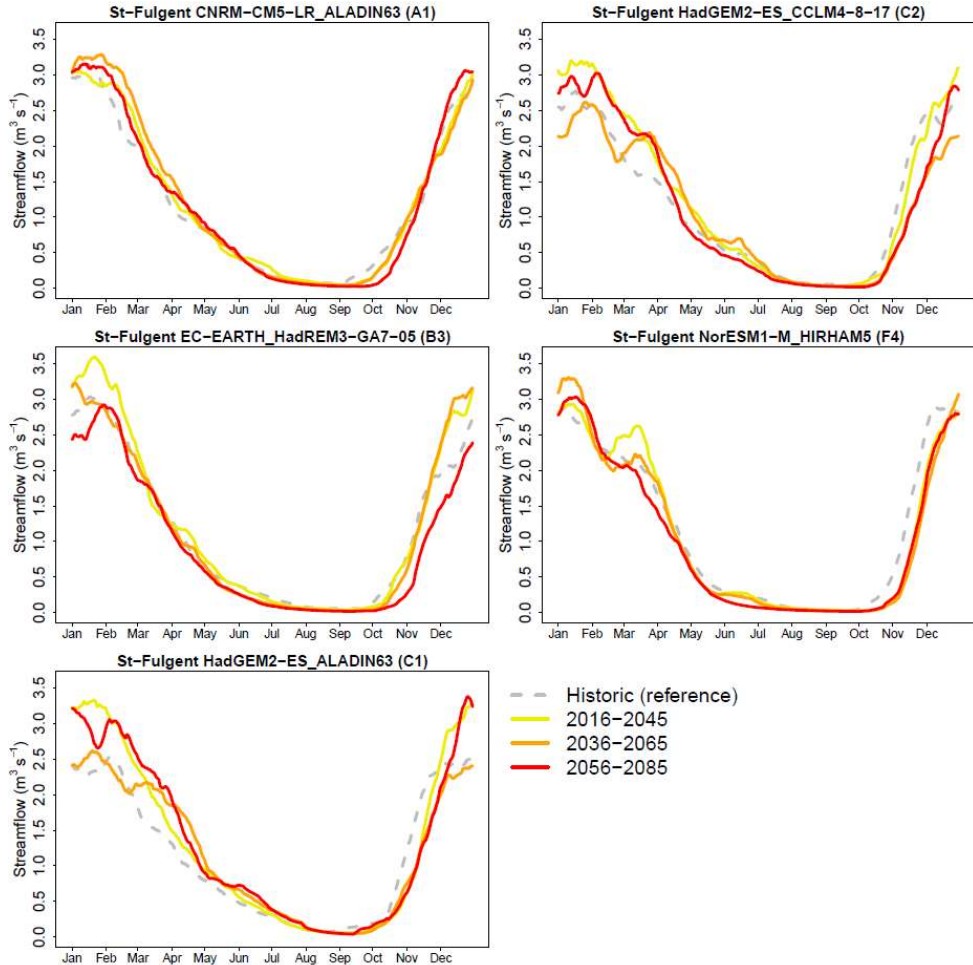

810

**Figure D2: Interannual streamflow regime for the projected uninfluenced streamflow for the St-Fulgent gauge station (upstream of the Bultière dam) for three future periods for the five climate projections.**