# Peer review of "Exploring future water resources and uses considering water demand scenarios and climate change for the French Sèvre Nantaise basin"

_EGUsphere, 2025_

## Author Comment (AC1)

We would like to express our gratitude to Dr Sopan Patil for his meticulous review. We provide below detailed answers (in black) to the remarks made by the reviewer (in blue). Line numbers and section numbers refer to those from the submitted manuscript.
* * *
Reviewer 1:

The Santos et al. manuscript presents an interesting study on the integrated water resource management (IWRM) modelling approach to explore the impacts of climate change and future demand scenarios at a catchment in Northwest France. While the overall scientific approach appears to be sound, I have a few comments and suggestions below, which will hopefully improve some aspects of the paper's presentation:

We thank the reviewer for his positive feedback.

1) The Methods section presents three versions of the hydrological model: calib, uninf, and iwrm. If I understood it correctly, only the calib version of the model is used for parameter calibration and model validation using the observed streamflow data, and the other model variants, uninf and iwrm, do not have the corresponding observed data to gauge their performance. However, Table 5 in the Results section shows the model performance of all three variants. So, what are the uninf and iwrm variants being compared against? And what is the point of showing which model variant performs best? As I understand it, each model variant serves a totally different purpose, and they are not competing against each other.

The reviewer is right, we do have three versions of the hydrological model, and the Calib version is the only one used for parameter calibration of the GR hydrological model. Since the available measured streamflow is, by definition, influenced by water uses in the catchment, the Calib version of the model incorporates observed withdrawals and releases (as described in section 2.2.3). By doing so, the simulated streamflow of the Calib version of the model is supposed to represent influenced streamflow. The uninf version of the model uses the same parameter set, but does not incorporate observed withdrawals and releases. Its objective is to dispose of a model version that simulates natural hydrology. It is therefore used to assess the impact of water uses on the Sèvre Nantaise hydrology (by comparing its output to a model version that incorporates water uses) and the projected evolution of natural hydrology. Finally, the iwrm version of the model still uses the same parameter set as for the other models, but incorporates water uses through the outputs of water demand models, and includes the management rules for deciding water withdrawal restrictions. This version of the model is necessary to simulate future projected influenced hydrology and water uses and assess the impact of water use scenarios and climate change on diverse indicators.

All three versions of the model simulate streamflow at the same stations, as a consequence the three of them can be evaluated against measured streamflow as performed in Table 5. By evaluating the three models against this same streamflow time series, we aim to i) show that the Calib version is sufficiently well calibrated, ii) illustrate the impact that not considering water uses can have on the performance of modelling (here by comparing the Uninf performance to the Calib performance) and iii) verify that the model version that uses water demand models (iwrm) performs reasonably well, before using it for projections. Therefore, we do believe that evaluating all three models against measured streamflow is totally meaningful.

We will however work on better justifying this approach and better introducing it. We will provide details on the KGE calculation, which compares observed data (i.e. influenced streamflow) with, respectively, simulated data from the "Calib" model (i.e. influenced simulated streamflow), from the "Uninf" model (i.e. uninfluenced simulated streamflow) and from the "iwrm" model (i.e. estimate of influenced simulated streamflow based on uses and management rules).

2) In my opinion, the water demand and release models of the iwrm variant are the most important contribution of this paper. However, more information might be needed to determine the robustness of the water demand and release models presented here. Not much information has been provided about the input data used for the models described in Appendix B. Where has this data been sourced from?

Water demand and release models indeed rely on numerous input variables. For instance, the cattle watering models rely on the number of heads for each cattle type, on the demand per head, on the source of withdrawal (natural environment or drinking water network) and on the daily partition over the year. The values of these variables for the reference period come from multiple sources, all in French, and most available by request only. There is unfortunately no database available, for privacy and economic reasons. For example, the number of heads was retrieved from local Agricultural chambers, the demand per head results from an estimation that was retrieved in a report from the SDGRE 49 (Schéma Départemental de Gestion de la Ressource en Eau du département 49), the source of withdrawal (natural environment or drinking water network) was provided for only two drinking water providers, therefore we had to extrapolate this information, and the daily partition over the year was retrieved from a report named « Étude sur la gestion quantitative de la ressource en eau en Bretagne. Analyse de la pression de prélèvement, définition des volumes disponibles, CACG, 2021 », which provided the monthly partition of annual water cattle consumption. As can be seen from this single example, the data sources of the inputs of the water demand and release models are very diverse, incomplete and sparse. An as complete as possible listing and description of these data sources, and the hypotheses made to address data gaps, is available in Santos et al. (2023a), in French. However, we do not believe that such information should appear in the present manuscript, because i) it could easily double the number of pages in the manuscript, ii) this is very specific to the Sèvre Nantaise catchment case: in another catchment, even another French one, different data sources might exist, different information could be available or missing and the actual figures would not necessarily be transferable to other areas. We believe that the actual added value of the Appendix B presenting the water demand and releases models are the equations, which can thereafter be applied using any available source of data. We understand that such a limitation prevents from any form of direct reproducibility in this part of our study, but this is not something we can avoid here.

Nonetheless, regarding this specific point, we will properly refer to Santos et al. (2023a) in the Appendix describing the Water demand and release models, to allow readers to find more detailed information about the data sources.

Reference:

Santos, L., Tales, E., Bluche, A., Thomas, A., Mounereau, L., Thirel, G. Etude HMUC : Rapport Phase 2. État des lieux / Diagnostic / Constitution de la modélisation. 197 p https://hal.inrae.fr/hal-04008873, 2023a.

Is the model implementation done in spatial grids, or is it spatially lumped at the subcatchment level?

We believe that this question applies to the water demand and release models. As written in the first line of the Tables of Appendix B, « Equation (are given) for location a and day d. Areas can be either municipalities or plots (for irrigation). The water demands are thereafter summed up over the diverse areas a of each hydrological modelling unit ». We will replace in this Appendix the word « areas » by « Locations » to avoid any misunderstanding, as it represents the same modelling unit. We will also replace "hydrological modelling unit" with "sub-catchment" to be consistent with the Material and Methods section. If the question applies to the hydrological model, as explained in section 2.3.1, we apply the GR6J model onto 32 sub-catchments in a semi-distributed manner.

The cattle watering model equation allows for information on different cattle types. How many different cattle types were considered? And how is their water demand calculated?

There are nine types of bovine cattle, two types of pork, sheep, goats, poultry, horses and rabbits, which makes it 16. The water demand is calculated thanks to the first two equations in Table B1. If the question related to the unit consumption, it was retrieved for each cattle type from the report from the SDGRE 49 mentioned in the previous answer and is available in Santos et al. (2023a). We will refer to this report but not detail all cattle types and unit consumptions in Appendix B for the sake of conciseness. Regarding the scenarios, the rate of evolution of the number of heads was considered for three groups of bovine cattle, one group for pork and poultry, and the number of heads for other cattle types was not modified, due to the low number of heads in the catchment.

The same question applies to the demand calculation for other uses.

We understand the rationale behind your question. We retrieved withdrawals for all main industries, which we used for 2008-2020. We will not answer into details to this demand but we will once again refer to Santos et al. (2023a).

Also, why does the formula for drinking water demand add cattle and industrial water demands to the population's consumption?

This is because a fraction of these water demands are made to the drinking water network (see the $R_{DW}(i, y)$ and $R_{DW2}(i, y)$ variables in the related equations).

Perhaps it might be useful to provide a detailed schematic, maybe at a subcatchment level, of how the different water demand models are feeding into the base hydrological model to create the iwrm variant.

Thank you for this interesting suggestion. Actually, the water demand and releases models do not interact with the base hydrological model internal variables, only with its simulated streamflow. We propose below a figure that explains the main water fluxes in the iwrm model version. In this figure, a catchment comprising an upstream subcatchment is shown, as it allows to show both a catchment with a dam reservoir, and a catchment with no dam reservoir. We hope this improves the understanding of the iwrm modelling. This figure could be inserted in section 2.3.2 Integrated water resources management modelling.

[Figure]

$Q_{tot} = (1-\alpha)Q_{sim_d} + lagged(Q_{dam} - Q_{Ind,NE} - Q_{vine} - Q_{CW,NE} - Q_{irrig}{}^* + Rel_{ww} + Rel_{DWloss} + Rel_{Ind,NE})$

*Figure: Schematic of the functioning of the integrated hydrological model. Two catchments are shown: the upstream catchment includes a dam reservoir (corresponding to MU1 and MU3 in the Sèvre Nantaise catchment), while the downstream catchment does not include any dam reservoir (corresponding to all other MUs). Small reservoirs and the related water demands, as well as wastewater treatment plants can be present in both cases but are shown in the downstream catchment only for graphical purposes. The dam reservoir outflow ($Q_{dam}$) is determined according to the management rules given in Table B3 and is delayed towards the catchment outlet proportionally to the hydraulic distance from the dam to the outlet. The simulated streamflow in the downstream catchment ($Q_{sim_d}$) is given by a GR6J model. Part of this ($\alpha$) is captured by small reservoirs. The total streamflow at the outlet is the sum of all streamflows (from the dam, the downstream catchment and the water releases) minus the water withdrawals, which are delayed according to the hydraulic distances between the withdrawal and release points. All other fluxes and notations are detailed in Appendix B. Natural fluxes are in dark green, streamflow is in blue, water withdrawals in light blue, and water releases in dark orange.*

3) Another potentially innovative aspect that has unfortunately been sidelined in the paper is information from the stakeholder workshops. I think more detailed information is needed on how the three future scenarios were initially designed and on the specific value added by the stakeholder feedback. As currently presented, we are only seeing the final product, and the importance of stakeholders in shaping these scenarios for the local conditions is being ignored.

Thank you for this suggestion. While our approach with stakeholder workshops used for designing the water use scenarios is definitely better than just designing scenarios between scientists, we do not see this approach as the most innovative aspect of our study. In other words, we believe that, although this approach is not yet that widespread, we did not "invent" anything. We also believe that what is replicable in other studies is already presented in the article, namely i) performing a literature review, followed by ii) the proposal of scenarios by scientists and finally iii) the discussion with stakeholders resulting in modification of the scenarios.

Globally, the scenarios that were initially proposed were neither rejected nor heavily modified. Some elements that emerged from the stakeholder workshops are (non-exhaustive list):

- The bird flu that was ongoing on the territory was not considered in the initial scenarios. As it seems to lead to a significant decrease of poultry cattle, the scenarios were modified to consider that;
- Vine spraying to prevent frost damage is an emerging issue in the catchment. The scenarios were modified to add this water demand;
- The need to consider specific practices, such as agroecology or the type of irrigation. The agronomic modelling cannot consider this and information about the current practices could not be provided by stakeholders, so the scenarios were not modified following these remarks;
- The trends of the evolution of populations were modified to better represent the local dynamics (see the evolutions in urban areas in the scenarios);
- The alternative scenario, previously named "adaptative" scenario, was renamed as some evolutions cannot be considered as adaptations;
- A newly planned inter-catchment water transfer was added.

We will add this information to the Appendix describing the scenarios.

4) Lastly, while the iwrm model presented in this study seems innovative, it is certainly not the first one to have attempted a quantification of future water demand. There have been a large number of studies conducted using other models, most notably WEAP, to address water demand management and forecasting. In this context, I find it troubling that the presented iwrm model, and its results, have not been discussed in the context of other existing models. It would be quite valuable for the authors to discuss the similarities and differences in the specific aspects of their iwrm model and others found in the literature.

We thank the reviewer for this suggestion. We will add a description of some iwrm approaches in the introduction, and we will also better discuss the results in regards to the literature.

---

## Author Comment (AC2)

* * *
Reviewer 2:

We would like to express our gratitude to Dr Joris Eekhout for his meticulous review. We provide below detailed answers (in black) to the reviewer's remarks (in blue). Line numbers and section numbers refer to those from the submitted manuscript.

The manuscript describes modelling study on the impact of climate change and water demand/supply scenarios on water availability in a French catchment. The authors analysed the current water supply and demand, based on existing data. A hydrological model was set-up and calibrated, considering water use in several sectors. The model was subsequently applied to a climate change scenario, considering 5 climate models. Moreover, 3 water use scenarios were defined and their impact assessed. The study shows that climate change will affect the water balance in the catchment, but that future water use does not differentiate that much between the different scenarios.

Overall, the manuscript is well-written and study is well-performed. However, there are some revisions needed. Especially the presentation of the results should be improved. It sometimes seems that I'm reading a data report, rather than a scientific article. The authors show so much results that is difficult, if not impossible, to extract the main message from each figure and table. Moreover, improvements to the Introduction (structure), Material & Methods (too much detail) and Discussion (too little detail) are needed.

We thank the reviewer for the detailed feedback on our manuscript.

The Introduction needs some revision. The most relevant concepts and previous research on the subject are described. However, the structure is not that clear. The authors suggest 2 options for studying water supply and demand in catchments. It is not clear what is actually the difference between the two options. Moreover, which of the two options is used in this study? If the two options are very relevant for the study, then this needs to be more integrated into the rest of the Introduction. For instance, the sentences prior to the objective should focus on these two options and the two options should be integrated into the objective as well.

We thank the reviewer for this remark. We agree that the distinction between the two options was not sufficiently clear. Following this remark and the remark from reviewer 1 to mention more clearly other iwrm approaches, we will delete the sentences about the two options, and orient this part of the introduction towards a better description of more classical approaches, such as WEAP and SWAT. We will also reorganize the introduction.

The Material & Methods section is rather long and provides maybe too much detail. Lines 139-143 and 194-199 give information that seems not too relevant for the study. There are also a few paragraphs that can be shortened without losing much relevant information, such as those paragraphs in lines 200-213. Many paragraphs start with a few (2-3) introductory sentences that can easily be replaced with a single sentence (e.g. lines 267-269 can be replaced by "Management rules were implemented in case of water deficit within the system.") Finally, I also suggest to move some figures and tables to the Supporting Material, such as Figure 2 and Tables 2 and 4.

Thank you for these suggestions. Lines 139-143 represent the real-life description of the functioning of the catchment, that is later on implemented in the model as described in lines 269-274. We will however shorten lines 139-143 as suggested, but we will keep part of it, as the reviewer later on asks based on what water restrictions are decided, which is what is explained here (see comment about lines 269-272). Regarding Lines 194-199, we agree, and recognized in the manuscript, that these are not central to the study. We initially added this information for transparency and because water use data are usually sparse in many areas, and knowing how the water use dataset was obtained can be of interest for readers; note that the remark 2) of the review of Dr. Sopan Patil is in line with this point of view. In order to shorten the main article, we propose to move these lines in a new Appendix, together with the paragraph in Lines 200-213, Table 4 and Figure 2, as also suggested. We will also modify Lines 267-269.

The authors use an ensemble of 5 climate models (GCM-RCM combinations) which is very important to account for climate model uncertainty. The authors show the results of each climate model individually, which leads to rather overwhelming figures with a lot of data, especially Figures 5-9. I highly suggest to show the ensemble average instead and determine the uncertainty of the results using an uncertainty band or use statistics to quantify uncertainty. This would make the figures much more interesting to look at. At the end, the readers will be interested in the general tendency of the results. The same holds for the results show for each individual management unit. This becomes much more appealing to look at if you show the results on a map. Readers are not going back and forth between the study area map and the results to see where runoff is going to increase or decrease. Readers are also not going to compare the results of all individual climate models. So, please reduce the amount of data shown in all these figures and try to provide enough data to make a coherent story.

We understand the comment from the reviewer regarding the use of the ensemble of five climate models. However, we would like to keep the present approach. The general approach we chose in this study relies on the storylines approach described by Shepherd et al. (2018). The general idea is, instead of using a large ensemble of projections, or even worse, selecting only one projection, to propose a selection of physically-consistent pathways. As argued by Sauquet et al. (2025, in review), who applied this approach to the climate projection dataset we used, and on which we based our work: "Unfortunately, uncertainties are sometimes ignored by stakeholders: For pragmatic reasons (limited computing resources) or because they do not know how to select a subset of projections, only one climate projection or median changes are considered in prospective studies. […] They do help stakeholders to make informed choices and illustrate climate-related uncertainties at the end of the century.".

The reviewer suggests showing both the average and an uncertainty band. We respectfully disagree with this suggestion, even though this is common practice in climate change impact studies. Indeed, both the average and the uncertainty bounds (represented by the min/max or some quantiles or other stats) all indicate values that do not represent a spatially-consistent simulation. Quite obviously, the average by definition mixes different projections, as also do quantiles. Regarding the uncertainty bounds, if they are represented by the min/max, they do represent single simulations, but considering them does not indicate whether the minimal value for one indicator corresponds to the minimal value for another indicator. Such representations lead to losses of information.

Our choice was made because the objective of this study was not a theoretical exercise on a given case study, but rather aimed at providing information to stakeholders or users that is fully meaningful and understood by them. We do believe that such an approach might develop in climate change impact

studies, as recommended by the IPCC, and as recommended by governmental and climate French institutions.

We understand that the amount of results provided is substantial. This matter of fact is inherent to the reality of a complex anthropized catchment and to its management, which are by definition very complex. If we want to provide explanations to the evolution of water uses in a catchment under climate change and water demand scenarios, it is necessary to account for spatial heterogeneities and to account for the different climate projections and scenarios. To help readers, we had tried to follow a strict colour code. Regarding the possibility to represent such results as maps, we propose as suggested to represent some of these figures as maps (Figure 6), as exemplified for Figure 6 below. Please note that we discarded Q05, Q50 and Q90 to simplify the results shown. For some other figures (Figure 7 and 9), we propose to provide aggregated results at the basin scale in the main body of the article (as exemplified for Figure 7 below), and to move the current representation to an Appendix, as we believe that both have interest. Figure 8 will now only include the catchment aggregated results (see justification below).

References:

Sauquet, E., Evin, G., Siauve, S., Aissat, R., Arnaud, P., Bérel, M., Bonneau, J., Branger, F., Caballero, Y., Colléoni, F., Ducharne, A., Gailhard, J., Habets, F., Hendrickx, F., Héraut, L., Hingray, B., Huang, P., Jaouen, T., Jeantet, A., Lanini, S., Le Lay, M., Magand, C., Mimeau, L., Monteil, C., Munier, S., Perrin, C., Robelin, O., Rousset, F., Soubeyroux, J.-M., Strohmenger, L., Thirel, G., Tocquer, F., Tramblay, Y., Vergnes, J.-P., and Vidal, J.-P.: A large transient multi-scenario multi-model ensemble of future streamflow and groundwater projections in France, EGUsphere [preprint], https://doi.org/10.5194/egusphere-2025-1788, 2025.

Shepherd TG, Boyd E, Calel RA, Chapman SC, Dessai S, Dima-West IM, Fowler HJ, James R, Maraun D, Martius O, Senior CA, Sobel AH, Stainforth DA, Tett SFB, Trenberth KE, van den Hurk BJJM, Watkins NW, Wilby RL, Zenghelis DA. Storylines: an alternative approach to representing uncertainty in physical aspects of climate change. Clim Change. 2018;151(3):555-571. https://doi.org/10.1007/s10584-018-2317-9.

[Figure]

Figure 6: Evolution in percentage of the uninfluenced hydrological indicators under five climate projections for the 11 Management Units of the Sèvre Nantaise for 2056-2085 compared to 1976-2005. The shaded bars indicate the -100 to +100 % range.

[Figure]

Figure 7: Relative evolution of water demand for the different uses (in lines) and scenarios (in columns) under five climate projections aggregated at the Sèvre Nantaise scale for 2056-2085 compared to 1976-2005.

The Discussion section needs to be significantly improved. Most discussion stays on the surface and does not go deep into the literature on how the results if this study compares to previous results. Neither do the concepts the authors refer to in the Discussion are backed by previous research. There is a lot of room for improvements in the Discussion.

Thank you for this comment. We will improve the discussion, using the suggestions that you proposed in your specific comments and those from reviewer 3, namely regarding other iwrm approaches, and other approaches such as decision scaling, scenario-neutral theory and info-gap theory.

The same can be said for the Conclusions. The Conclusions should be one-on-one related to the objectives of the study. I don't have the feeling that the Conclusions answer the objectives/research questions posed in the Introduction. Currently, it reads more like a summary, but includes also several lines about the implications of the study. So did the authors reach the objective of the study? And to what extent? Such questions should be answered in the Conclusions.

Thank you, we agree with this comment. We will improve the conclusions.

Below I have provided specific comments to the text, figures and tables.

Thank you again for your thorough review of the manuscript.

Specific comments

Lines 19-22: The authors refer here to water demand satisfaction, but it is unclear what is meant by that. Can this be quantified? I guess this must be the case, because the authors use a water resources model in their study. It would be useful if the authors can give quantified estimates of the water demand satisfaction or give a clear definition what is meant by it.

The reviewer is right, we did not properly define this concept. Water demand satisfaction simply represents the discrepancy between the water demand (i.e. how much water users would like to withdraw) and the actual water withdrawal (i.e. how much could actually be withdrawn due to lack of water resources, or restrictions, for instance). In our case (section 3.5), we express water demand satisfaction as the ratio in percentage between water withdrawal and water demand.

We propose to define water demand satisfaction when the term is first used (in the abstract and in the introduction), and to define our calculation in this study in section 2.3.3 when the concept is mentioned.

Lines 19-20: Please clarify in the text what you mean with "A single climate projection" and "a less drastic deterioration of the system".

This sentence is indeed not that clear. We propose the following: « Only one climate projection does not indicate a strong decline of low flows and water demand satisfaction, and this is only the case in some parts of the catchment. »

Lines 41-47: It is unclear what the main message is of this paragraph. It seems somewhat directed to the fact that hydrology can be studied with models. I guess this can be included in a single sentence to start the next paragraph.

Thank you. We propose to delete this paragraph and insert the following sentence at the beginning of the following paragraph:

« Numerous hydrological models, each based on different assumptions, exist (Hrachowitz and Clark, 2017). Despite the diversity of approaches, and their diverse qualities, when attempting to comprehend and depict anthropized catchments, it becomes clear that relying solely on hydrological models is insufficient. ».

Lines 51-52: You mean water supply and water demand studies? Please clarify in the text.

Due to modifications of the introduction, this sentence will not exist anymore.

Line 51: "This approach...with demand". This part of the sentence largely repeats the previous sentence. Please remove or revise.

The reviewer is right, however, due to modifications of the introduction, this sentence will not exist anymore.

Lines 52-53: Are hydrological models really needed to estimate water supply and water demand? From the previous sentences it does not become clear why that is the case. Please clarify.

Hydrological models are necessary to estimate the natural water resources: if streamflow observations are available, these are impacted by water use and cannot be compared to water use data. Due to modifications of the introduction, this sentence will not exist anymore.

Line 55: What is meant by coarse models? Please clarify in the text.

We meant « simple ». Due to modifications of the introduction, this sentence will not exist anymore.

Lines 56-60: It is difficult to understand the difference between the two approaches, because both seem to involve models. So what is the actual difference? Please clarify this in the text.

Due to modifications of the introduction, this sentence will not exist anymore

Lines 67-68: It seems from this sentence that integrated and sustainable water resources management is a concern, but I suspect that the lack of management will be a concern, not the management itself. Please revise this sentence.

We propose the following rewording:

« In France, it is identified by law as a key concern the fact that water resources must be managed in an integrated and sustainable way. »

Lines 76-77: This refers to the first option from lines 50-55? In that case, say first option or give this option a specific name in lines 50-55 and refer to that name. This would clarify this for the readers.

Due to modifications of the introduction, this sentence will not exist anymore

Lines 67-81: These sentences put the rest of the Introduction into a local perspective, which is fine. However, there is quite some repetition in these sentences and this local focus is maybe better suited to go into detail under Material & Methods and Study Area. I suggest to shorten this part of the Introduction to a few sentences in which the main points of the Introduction are summarized from which follows the objective of the study. So what is the research gap the authors are focusing on?

We thank the reviewer for this remark. We will reduce this part of the Introduction and better introduce the research gap we focus on, namely the development of an integrated water resources management modelling.

Lines 85-88: These specific objectives do not add much to the two main objectives. I suggest to replace these sentences with a few (2-4) sentences in which the methods are briefly explained.

We propose to simply delete these sentences.

Line 94: Replace "who" with "which".

Thanks.

Figure 1: The legend says that with blue the rivers and dams are indicated, but it is in fact the rivers and the reservoirs. It would be useful to indicate the dams as well. I'm not sure if it is necessary to indicate the management units and the sub-catchments. The location of the sub-catchments might be arbitrary or is there a physical reason for their location?

Regarding the blue colour in the legend, we will modify to « dam reservoirs », as simply writing reservoirs would be incorrect, as many small reservoirs (> 11000) are spread over the catchment.

Regarding the location of the sub-catchments, as written in the Figure 1 caption, we refer the reader to section 2.3.1, where we indicate: « The subdivision of the catchment into 32 sub-catchments was designed to align with locations of the 13 gauge stations (listed in Table 2) and the 11 management units (MUs) used for water management in the Sèvre Nantaise (Figure 1). » In addition, we refined some sub-catchments to obtain areas with relatively similar surfaces. We will add the last sentence in section 2.3.1. We believe that it is necessary to keep the Management Units in this map, as this is the spatial unit at which natural and anthropized fluxes are located thereafter. It allows to localize the main stocks and cities, which are thereafter mentioned.

We will remove the sub-catchments as suggested but we will keep the management units.

Lines 108-109: What is considered upstream and downstream? Please clarify in the text.

Upstream is the south-east part of the catchment, and downstream is the Nantes area. We will rephrase.

Table 1: I suppose this table does not contain the data for the large reservoirs. Then there must be an upper limit for the capacity of the third category. Please indicate this in the table.

Yes indeed, as written in the Table 1 caption, these data are for the small reservoirs. The upper limit is 165 000 m², we will add this information in Table 1.

Lines 166-168: Why did the authors consider the RCP8.5 scenario? Please clarify in the text.

The RCP8.5 is a scenario that is very often used in the literature, including in HESS papers, and its choice is rarely justified. In our case, we made this choice because this scenario was, at the time, the most likely scenario in terms of temperature increase, the scenario with the largest amount of available projections and the largest amount of feedback. We will clarify.

In addition, the French government asked, after our choice of projections, that all works about climate change adaptation must target a future climate for which the future air temperature over France increases by + 4 °C compared to the pre-industrial period. Such temperature increases can only be found in the RCP 8.5 projections (Corre et al., 2025). This is an additional, although a posteriori, justification of our choice, which we will not mention in the manuscript.

Corre, L., Ribes, A., Bernus, S., Drouin, A., Morin, S., Soubeyroux, J.-M. Using regional warming levels to describe future climate change for services and adaptation: Application to the French reference trajectory for adaptation. Climate Services 38, 100553. https://doi.org/10.1016/j.cliser.2025.100553, 2025.

Lines 168: Please explain briefly how the ADAMONT bias-correction method works.

The ADAMONT bias correction method is a variant of the quantile-mapping approach that forces the statistical distribution of the simulated atmospheric variables to match that of the SAFRAN atmospheric reanalysis. We will add this information.

Lines 168-172: The two criteria used to select the 5 GCM/RCM combinations should be better described. How did you compare the EURO-CORDEX projections with the CMIP6 projections? What do you consider representative with respect to the RCP8.5 scenario at the end of the 21st century? Please clarify these issues in the text.

The first criterion was in fact directly applied by Marson et al. (2024) in order to exclude climate projections for which the seasonal precipitation and temperature evolution (future period versus reference period) were outside the quantile 5 % / quantile 95 % range of CMIP6 projections over France (see Figure 4 of that report).

The second criterion was recommended by Marson et al. (2024), but because they applied it at the France-wide scale, we did not rely solely only on their selection but we rather applied it ourselves at the Sèvre Nantaise catchment scale. Still, we have 4 of our 5 projections common with Marson et al. (2014).

The general approach relies on the storylines approach described by Shepherd et al. (2018). The general idea is, instead of using a large ensemble of projections, or even worse, selecting only one projection, to propose a selection of physically-consistent pathways. To do so, a rather qualitative selection based on an expert-wise subjective analysis is performed. No calculation with thresholds to respect is done; only the fact that these projections are relatively well spread in the figures of the Appendix A of our manuscript is considered, similarly to Marson et al. (2024)'s approach.

We propose the following rewording:

« The selection of the five GCM/RCM projections followed the recommendations of Marson et al. (2024). First, we relied on the dismissal by Marson et al. (2024) of climate projections whose seasonal precipitation and air temperature evolutions were outside the Q5/Q95 range of CMIP6 projections

over France. CMIP6-based projections could not be used as they had not yet been regionalised over France. Second, we adopted the storyline approach introduced by Shepherd et al. (2018) and also selected recently for climate projections over France by Marson et al. (2024) and Sauquet et al. (2025). The storyline approach relies, instead of using a large ensemble of projections and assessing climate-related evolutions based on probabilities, on a selection of physically-consistent pathways. Such storylines must be adapted to the study objectives. Consequently, we selected five projections whose seasonal precipitation and air temperature evolutions were contrasted, among the available projections of the Explore2 dataset. »

References:

Marson, P., Corre, L., Soubeyroux, J.-M., Sauquet, E., "Rapport de synthèse sur les projections climatiques régionalisées", https://doi.org/10.57745/PUR7ML, Recherche Data Gouv, V1, 2024.

Sauquet, E., Evin, G., Siauve, S., Aissat, R., Arnaud, P., Bérel, M., Bonneau, J., Branger, F., Caballero, Y., Colléoni, F., Ducharne, A., Gailhard, J., Habets, F., Hendrickx, F., Héraut, L., Hingray, B., Huang, P., Jaouen, T., Jeantet, A., Lanini, S., Le Lay, M., Magand, C., Mimeau, L., Monteil, C., Munier, S., Perrin, C., Robelin, O., Rousset, F., Soubeyroux, J.-M., Strohmenger, L., Thirel, G., Tocquer, F., Tramblay, Y., Vergnes, J.-P., and Vidal, J.-P.: A large transient multi-scenario multi-model ensemble of future streamflow and groundwater projections in France, EGUsphere [preprint], https://doi.org/10.5194/egusphere-2025-1788, 2025.

Shepherd TG, Boyd E, Calel RA, Chapman SC, Dessai S, Dima-West IM, Fowler HJ, James R, Maraun D, Martius O, Senior CA, Sobel AH, Stainforth DA, Tett SFB, Trenberth KE, van den Hurk BJJM, Watkins NW, Wilby RL, Zenghelis DA. Storylines: an alternative approach to representing uncertainty in physical aspects of climate change. Clim Change. 2018;151(3):555-571. htps://doi.org/10.1007/s10584-018-2317-9.

Line 179: Replace "to an increase" with "in an increase".

Thanks.

Table 3: I suggest to replace the symbols (+, -, =) with the actual change in percentage for precipitation and degrees for temperature. Or otherwise, explain what the symbols mean. But my preference goes to quantified estimates of the change in precipitation and temperature.

Thanks for this comment, we will replace the symbols with the actual numbers. Please note that these numbers are also graphically represented in the figures of Appendix A.

Line 191: With "heads" the authors mean "animals"?

Yes, the number of animals are generally counted in « heads » (see e.g. this USDA document https://www.nass.usda.gov/Publications/Highlights/2024/Census22_HL_Cattle%20and%20Cattle%20on%20Feed_final.pdf or this FAO report https://www.fao.org/4/i3138e/i3138e07.pdf).

Lines 191-193: Please indicate how many years data are available for irrigation and drinking water withdrawals. It seems a bit unclear now how much data are available.

Daily data for the Bultière drinking water were available from 2008 to 2020, and from 2010 to 2020 for the Ribou-Verdon dam reservoir. For irrigation, seasonal data for five years were available only for few sectors.

We believe that this information is not primordial for the manuscript. For the sake of conciseness, we prefer not to include this information in the revised manuscript, and rather refer to the report.

Line 227: Please revise this sentence and explain the parameters in a logical order, i.e. "X1 is the production capacity parameter (mm), X2 is the inter-catchment exchange coefficient (mm d-1), X3 is...".

We are not quite sure what the reviewer means with logical order, do you mean respecting the numbering of the parameter names? We will do so. We just want to inform the reviewer that this ordering is actually not "logical", as it does not follow the actual water fluxes within the model.

Lines 235-237: I guess each sub-catchment is considered a homogeneous unit within the model. How does land use and soil characteristics affect the behaviour of the model? Should each sub-catchment not be homogeneous considering land use and soil? Please explain in the text.

Indeed, each of the 32 sub-catchments is simulated by a single GR6J occurrence, meaning that the meteorological data is spatially aggregated over each sub-catchment, and that we assume that the processes over the sub-catchment are homogeneous. Land use and soil characteristics of the sub-catchments do affect the processes and therefore the model behaviour, in a different way for each sub-catchment. This is however not taken into account explicitly, i.e. by actually considering land use and soil characteristics within the model, but rather through the optimisation of the model parameters, which are adapted, thanks to the optimisation algorithm, to fit the precipitation – discharge relationship in a different way within each sub-catchment. As this is a classical use of rainfall-runoff models as is done in the literature for decades, we do not feel that it must be specified in the manuscript.

Lines 244-246: So the authors applied the GR6J model using the airGR R package? Please indicate this in the previous subsection.

We will do so, thank you.

To clarify, we used airGRiwrm, which uses airGR itself regarding the hydrological modelling part.

Lines 269-272: Can the levels of restriction be quantified? Are they related to the discharge or total water availability? Please clarify in the text.

Regarding the first question, we are not sure we understand it. Indeed, the levels of restriction, in the sense of the amount to which water uses are restricted, are specifically mentioned in the lines 269-272.

Regarding what the restrictions are based on, as was specified previously (see lines 141-143: "In the event of a water shortage, local authorities are empowered to impose a series of restrictions across sub-catchments. These measures are primarily based on streamflow observations from predefined

gauging stations, while also considering local conditions, which may vary over time and space."), restrictions are mainly related to streamflow levels, yes. Despite the earlier demand of the reviewer to delete lines 141-143, based on this second request, we will keep them, and only slightly modify the text.

**Line 270: With "interdiction" the authors mean "suppression"?**

Yes, thank you.

**Line 306-308: This was indicated already a few lines back, please remove.**

We disagree, the lines 292-295 are about calibration only, not evaluation. Here, for evaluation, in addition to the KGE Box-Cox, we also provide the KGE as well as the bias (and formerly the correlation, but we will delete it based on another comment of the reviewer). Consequently, we will keep these lines.

**Line 335: Why is this a remarkable result? Please clarify in the text.**

We understand this comment as a question about why the Tillières station shows the lowest KGE values across all three models. To be honest, we do not have a clear explanation for that: that might be due to the small catchment size, or to data of lower quality. Despite the lower performance, it remains reasonably good. We will remove this sentence to avoid unnecessary disturbance.

**Line 346: I'm not familiar with using the correlation (between observed and simulations?) as a model performance metric in hydrological modelling studies. Given the low impact on the results, I suggest to remove this from the manuscript.**

Yes, as with any performance metric, the correlation is calculated by comparing observed and simulated streamflow. We are surprised that the reviewer is not familiar with this metric for hydrological modelling, as this is a component of the wide-spread KGE criterion (see e.g. a recent use in Munoz-Castro et al., 2025). It can be for example very informative for the timing of high flows or more broadly for the dynamics. We agree that it is not central to this study, so we will remove it.

Reference:

Muñoz-Castro, E., Anderson, B. J., Astagneau, P. C., Swain, D. L., Mendoza, P. A., and Brunner, M. I.: How well do hydrological models simulate streamflow extremes and drought-to-flood transitions?, EGUsphere [preprint], https://doi.org/10.5194/egusphere-2025-781, 2025.

**Lines 363-370: It is perfectly fine to remove this paragraph and first sentence of the next paragraph, which mostly repeats what is already included in the Material & Methods section.**

We agree that lines 363-368 mostly repeat the Material and Methods section. We included these sentences to make sure that the reader knew exactly what was done in this section. We hope that the rewording of the Material and Methods section (see answer to the first comment of reviewer 1) will

help in this regard, and we will remove these lines. We will keep lines 369-370 as this information is not given in the Material and Methods section.

We mean that the water management and planning is performed at the scale of the Management Units (MUs). We will remove this element here, as requested by the reviewer in the previous comment, and include that in section 2.1.3.

Figure 5: Here the authors show the results of 2 future climate projections that were not mentioned in the Material & Methods and are neither included in all other figures. I suggest to remove these two projections and solely focus on 2056-2085, this would make the manuscript much clearer to follow.

We agree that the two time periods shown in this figure are not properly mentioned in the Material and Methods section, and are not central to this study, so we will remove them from this figure (see below), as well as in the Appendix showing results for Cholet and Saint-Fulgent.

[Figure]

401: Please delete this sentence, it repeats the title of this subsection.

Ok.

Line 410-412: It seems that irrigation water demand is increasing in all three scenarios, also for the alternative scenario. So why do the authors claim that the alternative scenario leads to an overall decrease? Please clarify in the text.

The reviewer is right, in that sentence we meant that relatively to the other two scenarios, the water demand for the alternative scenario is lower. We propose the following rephrasing: "We find that the alternative scenario leads to overall lower or similar water demand compared to the constant and trend scenarios, while the trend scenario leads to the largest water demand, except for cattle watering, for which the current decrease is extended."

Lines 412-414: I don't see much differences between the MUs. Like argued before, try to simplify the results and show general tendencies, especially when there are no differences between MUs and between climate models.

We do agree that the differences between some MUs are minor for the water demand, because even if there are specificities in the different MUs, we applied rather generic percentages (see Appendix C). However, we still see some differences, with MUs that have drinking water but no industry, MUs that have both, MUs that have none, and MUs that have industry but no drinking water. In addition, this only applies to water demand: the evolution of the uninfluenced hydrological indicators (Figure 6) shows some diversity, and the evolution of water demand satisfaction shows even more (Figure 9). For these reasons, and to ease to the comparison between MUs and the understanding of the different behaviours of MUs, we believe that we should keep all MUs.

In addition, we gathered a very detailed database of uses, and we set up a semi-distributed hydrological model accounting for water uses over several hundreds of points, aggregating all these results and losing the spatial diversity over the catchment would be a pity and a loss of information for this study.

Figure 7: The order of the scenario is consistent with the description of the scenarios in lines 326-327, i.e. constant, trend, alternative. Please adjust the order according to the Material & Methods.

Thank you, we will do that for Figure 7, but also for Figure 9 (see below), that will actually be moved to a new Appendix.

[Figure]

Above, the new Figure 7 that will be move to the Appendix.

[Figure]

Above, the new Figure 9 that will be moved to the Appendix.

Ok.

Lines 429-438: These sentences are very hard to follow because of the focus simultaneously on MUs and scenarios. Please revise.

These sentences have been almost completely removed, due to other changes on the related figure.

Figure 8: I'm really lost at this point. The terms "influenced indicator" and "uninfluenced indicator" only appear in this caption. There is an uninfluenced model, does this refer to the uninfluenced indicator? I had to look up that QA refers to the mean streamflow, why would an abbreviation be needed here? The same holds for the low flow indicator. I suggest to use mean flow and low flow instead of QA and QMNA5. Apart from that. Is it not more logical to present these results as a change from the reference scenario (uninfluenced?) to the scenarios with climate change (influenced?)? The mean flow seems not to change, that seems odd.

Yes, "uninfluenced indicator" refers to an indicator that is calculated from simulations of the uninfluenced model (Uninf). This term was already used in Figure 6. We will however define it in the caption of Figures 6 and 8, but will keep this short terminology inside Figure 8.

QA was properly introduced in line 310 and was also used in Figure 6 beforehand. Mean flow and low flows could be defined with many indicators, providing the name of the indicator is more rigorous. We will however mention the meaning of QA (and QMNA5) in the Figure captions in the revised version of the manuscript.

The rationale behind this representation was to be able to present relative changes between periods (as is done in the rest of the manuscript) but in the same time to compare the indicators of the influenced model with those of the uninfluenced model. The reviewer suggests to "present these results as a change from the reference scenario (uninfluenced?) to the scenarios with climate change (influenced?)". We think that there is a misunderstanding: actually, the "reference scenario" is a term that we did not use in the manuscript, we rather used the term "reference period", which corresponds to 1976-2005. For this reference period, we do have the inclusion of influences for the "iwrm" model, and we have no influences for the "Uninf" model, which allows to compare comparable things together. To improve the understanding of the present analysis, and to comply with the other general remarks about the representations we had proposed, we propose to make the following changes:

- The evolution of QA and QMNA5, i.e. the indicators representing, respectively, the mean flows and the low flows, between the future period and the reference period, will be presented in the main body of the manuscript for the outlet of the catchment, i.e. at Nantes (MU9) (see below). This figure shows that QA is not impacted by the scenarios of even by the water uses, but only by the climate projections.

[Figure]

*Figure 1: Evolution of mean flows (QA, row 1) and low flows (QMNA5, row 2) indicators at the catchment outlet (Sèvre Nantaise at Nantes) for the uninfluenced simulation and the three future scenarios for the five climate projections. The evolutions are calculated from 2056-2085 to 1976-2005.*

The fact that the mean flow seems not to change is not surprising. Actually, it does change, but very slightly, making it not visible in the figures. This is mentioned in lines 424-425 ("…with only minor influence from the water demand scenarios or the consideration of water use in general. This means that water consumption is not large enough to alter the total amount of water flowing into the rivers at the MU scale."). We can make a coarse and rapid calculation to explain that. At the total catchment scale, during the 2008-2020 period, the annual withdrawn volume is around 30 Mm$^3$, and the annual released volume around 22 Mm$^3$, implying an annual water consumption of 8 Mm$^3$. The mean annual streamflow of the Sèvre Nantaise outlet at Nantes is 22.76 m$^3$ s$^{-1}$ (see Table 1), which can be converted in m$^3$: 22.76*60*60*24*365.25 = 718 250 976 m$^3$ = around 718 Mm$^3$. The annual water consumption therefore represents around 1 % of the total streamflow flowing in the river at the catchment scale. Although we did not calculate actual numbers, it is easy to assess that the differences between the scenarios only represent a part of the total water consumption. Therefore, the difference between the scenarios regarding the volumes annually consumed is logically much smaller than 1 % of the annual streamflow, which is reflected by the fact that the mean flow indicator seems not to change in Figure 8.

Lines 473-484: The first part of this paragraph focusses on the climate change results, while the second part focuses on the scenarios. I suggest to separate these two discussions.

We will separate these two parts into two paragraphs.

Lines 475-478: There is no need to repeat these details in the Discussion section. Please focus on the discussion of the results. Please remove.

We will shorten these sentences.

Lines 478-479: Most readers are unfamiliar with the Explore2 project. So in what sense are the results coherent with this project? Please clarify in the text.

They are coherent with the Explore2 project results in that for this area, most of projections predict a strong decrease of low flows, and a likely decrease of mean and high flows. We will clarify.

Lines 485-508: I'm not sure what point the authors try to make here. So the scenarios do not show much differences between each other, apart for certain climate models. The latter should not be included in the discussion, because of the high uncertainty in the climate model output. Again, the average climate model ensemble should be considered, instead of the individual climate models. Regarding the scenarios. The information provided in the appendix that shows the differences between the scenarios does not allow to compare the different scenarios so easily with each other. It would be better to present 1 table in which all three scenarios are shown next to each other. So I cannot judge if the scenarios are that different from each other. It seems from lines 496-498 that the

authors already expected that the scenarios were not so different from each other, so why not focus on only 1 scenario? How did other studies deal with this?

We thank the reviewer for raising this important point about the difficulty of using several climate change projections, especially when the uncertainty is large. We must remind here that this work is not a fundamental research exercise in which we can afford limiting our results to probabilities. This work is also not a study in which numbers about future evolution of indicators must be produced without considering uncertainties. This work is based on a real-life need, with real-life decisions that will be based on the results. This means that uncertainty, which is real, cannot be ignored, even if this makes the message about the evolutions less clear. Stakeholders must make decisions under uncertain conditions! Also, while the storylines represent different physically-plausible scenarios, the ensemble mean of climate projections is a scenario that is very unlikely to happen. In addition, the ensemble mean is a scenario that smooths extremes and that the variability of the ensemble mean is much less than individual models or observations (Gleckler et al., 2008) and so does not represent a potentially real climate (Abramovitz et al., 2014). These references will be added in the manuscript. For all these reasons, and based on the state-of-the-art literature about storylines, and the guidelines from Explore2 (see Sauquet et al., 2025), we want to keep our five projections. In addition, we do not wish to exclude the climate projections from the discussions as suggested by the reviewer, as this is the very part where we explain the expected evolutions of water demand and satisfaction conjointly with climate evolutions.

We agree that the presentation of the scenarios could benefit from gathering all evolutions in a single table. Following this suggestion, and the fact that the 2016-2045 and 2036-2065 periods are excluded from the next version of the manuscript, we propose the following table, that focuses on the evolution of parameters in the 2056-2085 period compared to the reference period. Please note that for some parameters, the evolution is progressive, therefore we kept a progressive description of their evolution.

| Item | Spatial scale | Unit | Reference | Scenario | | |
|------|---------------|------|-----------|----------|------|------|
| | | | | Constant | Trend | Alternative |
| Drinking water and sewage treatment | | | | | | |
| Population | Heterogeneous over the catchment | Percentage of evolution | 2008-2020 | Unchanged | From 0 to +0.5 % between 2023 and 2059 following a linear trend. Stable after 2059. Main cities follow a higher increase (+0.1 %), while rural areas follow a lower increase (-0.1 %) | From 0 to +0.5 % between 2023 and 2059 following a linear trend. Stable after 2059. Main cities follow a higher increase (+0.2 %), while rural areas follow a lower increase (-0.2 %) |
| Consumption per unit | | | | Unchanged | Stable overall, but +50 L/inh./d for rural areas, - | Stable overall, then -1 % per year |

| | | | | | 50 L/inh./d for urban areas | from 2040 to 2050 |
|---|---|---|---|---|---|---|
| Drinking water network efficiency | Whole catchment | | | Unchanged | +0.013 % per year | +0.08 % per year |
| Partition between collective and individual sewage treatment | | Rate | | Unchanged | Incoming population considered as within the collective network | |
| Inter-catchment transfers | | - | | Unchanged | Unchanged except if already planned | |
| **Agriculture** | | | | | | |
| Dairy cows | Whole catchment | Percentage of evolution of the number of heads | 2008-2020 | Unchanged | -0.6 % per year | |
| Suckler cows | | | | Unchanged | -2.0 % per year | -1.4 % per year |
| Calves | | | | Unchanged | -0.7 % per year | |
| Porks | | | | Unchanged | -0.4 % per year | -0.5 % per year |
| Poultry | | | | Unchanged | -2.0 % per year | -1.4 % per year |
| Consumption per unit | | L d$^{-1}$ head$^{-1}$ | | Unchanged | Unchanged | Unchanged (except during days with air temperature > 30 °C) |
| Part of water withdrawal coming from the drinking water network | | Rate | | Unchanged | Unchanged | +20 % |
| Total cultivated area | | Percentage of evolution | | Unchanged | -0.2 % per year | Unchanged |
| Crop rotation | | Type of crop | | Unchanged | 10 % reduction over 10 years in vineyards and forage crops, replaced by wheat, maize, rapeseed and market gardening (in equal proportions, depending on what was already present) | Wheat is partly replaced by barley, and maize by sorghum, at a rate of 0.5 % per year. |
| Vine water spraying | | - | | Unchanged | On half of vineyards | |
| Irrigation practices | | - | | Unchanged | Unchanged | |
| Irrigated surfaces | | km² | | Unchanged | +15 % | |
| **Industries** | | | | | | |
| Industrial activity | Whole catchment | Percentage of evolution | 2008-2020 | Unchanged | +3 % | |
| Consumption per unit | | | | Unchanged | -6 % | -8 % |

Regarding the choice of showing 3 scenarios, even though we expected that their effect would be limited compared to the uncertainty of climate change, we believe that this is an important result to show. This is important to show the effect of the constant scenario, because that shows what to expect in the future if we freeze our water uses. This is important to show the effect of the trend scenario,

because it shows what to expect if we continue in a business-as-usual behaviour. And this is important to show the alternative scenario because it shows that modifying water uses does have an impact (although not enough to compensate climate change). More globally, due to the important uncertainty on the evolution of water demand, it is clear that using several scenarios is primordial. This is somehow a similar approach to that of RCPs or SSPs, where multiple socio-economic scenarios are proposed. Beck and Bernauer (2011) use 3 water demand scenarios, Zhang et al. (2023) use 6 water demand scenarios, for example. These references will be added in the manuscript.

References:

Abramowitz, G., Herger, N., Gutmann, E., Hammerling, D., Knutti, R., Leduc, M., Lorenz, R., Pincus, R., Schmidt, G.A., 2018. Model dependence in multi-model climate ensembles: weighting, sub-selection and out-of-sample testing. https://doi.org/10.5194/esd-2018-51

Beck, L., Bernauer, T., 2011. How will combined changes in water demand and climate affect water availability in the Zambezi river basin? Global Environmental Change, Symposium on Social Theory and the Environment in the New World (dis)Order 21, 1061–1072. https://doi.org/10.1016/j.gloenvcha.2011.04.001

Gleckler, P.J., Taylor, K.E., Doutriaux, C., 2008. Performance metrics for climate models. Journal of Geophysical Research: Atmospheres 113. https://doi.org/10.1029/2007JD008972

Zhang, Z., Getahun, E., Mu, M., Chandrasekaran, S., 2023. Water Supply Planning Considering Uncertainties in Future Water Demand and Climate: A Case Study in an Illinois Watershed. JAWRA Journal of the American Water Resources Association 59, 449–465. https://doi.org/10.1111/1752-1688.12948

Lines 509-532: This subsection gives very little discussion on what I would expect from the title. I was expecting more in-depth discussion on the model structure and how to incorporate water resources data into these models. The presented discussion stays very much on the surface, without going deeper into the literature on how previous studies dealt with these issues. Please revise.

Thank you for this comment. We will reduce the existing text, add some references and better discuss our approach regarding the different items, and we will also add a discussion about the practical implementation of the iwrm model.

Lines 534-555: Obviously, the authors should assess the climate model uncertainty. Similar to the previous subsection, this discussion stays on the surface. It highlights the uncertainties that are at play, but does not discuss which uncertainties are more important in the current study and how that compares to previous research. Please revise.

We are rather puzzled by this comment.

Firstly, we believe that this discussion currently goes further than just stating that we should assess climate model uncertainty. We present another method (ANOVA), that could be used in certain cases and we justify why we did not use it. We also do not limit this discussion to climate model uncertainty.

In addition, the "ranking" of uncertainties is clearly described in this section, see: "Nevertheless, the present study helps identifying some of the primary drivers of uncertainty in water resources, water demand, influenced streamflow, and water demand satisfaction evolutions, as discussed in the

previous sections. Specifically, climate projections appear to be the dominant source of uncertainty for natural water resources, influenced streamflow, water use satisfaction, and irrigation water demand evolutions. Water use scenarios introduce a certain degree of uncertainty regarding water demand, influenced streamflow, and water use satisfaction evolutions, albeit generally at a lower level than climate projections. (…) It should be noted that the uncertainty related to greenhouse gas emission scenarios and hydrological modelling was not assessed in this study for the sake of simplicity. However, previous studies have shown that these modelling steps can constitute a substantial source of uncertainty, especially for low flows (Vidal et al., 2016)."

We are sorry if we misunderstood the message in this comment.

Reference:

Vidal, J.-P., Hingray, B., Magand, C., Sauquet, E., and Ducharne, A.: Hierarchy of climate and hydrological uncertainties in transient low-flow projections, Hydrol. Earth Syst. Sci., 20, 3651–3672, https://doi.org/10.5194/hess-20-3651-2016, 2016.

Lines 557-562: Please reduce the description of the methods to a single sentence and focus on the conclusions of the study.

Ok.

Lines 562-563: Please remove this sentence.

Ok.

Lines 567-568: I'm not sure to which scales the authors are referring to in this sentence. What is the catchment-scale in this case? The whole catchment or the MUs? What would be a larger scale? The authors did not perform analysis at larger scales, so how can they be so sure that at larger scales these processes are not becoming visible? Please clarify in the text.

The catchment scale mentioned here is the Sèvre Nantaise total catchment scale, otherwise we would have used the terms sub-catchment scale or MU scale. We checked the whole manuscript and our use of this term is consistent. Larger scales are usually continental or global scales. We will add this information. We are pretty sure that global-scale studies do not account for the > 11,000 small reservoirs of the Sèvre Nantaise, that they do not consider the actual streamflow thresholds used by local authorities to restrict water uses, and that the water demand scenarios do not consider local trends such as the ones we included by discussing with stakeholders. We will add a couple of references using classical IWRM approaches on larger areas than the Sèvre Nantaise, and which only consider a hundred of reservoirs or even less.

Lines 569-575: These are more implications of the study and go beyond the current study. This would be better fit for the Discussion section.

Closing the conclusions with an opening related to potential implications of the work is rather classical, so we prefer to keep these sentences and not to dedicate a specific discussion related to this point.